# PH-SHOWOA: Parallel hybrid SHO-WOA for VRPSPDTW

**Tram Nguyen**[1], **Snasel Vaclav**[2], **Bay Vo**[3], **Van Du Nguyen**[1]*

**1** Faculty of Information Technology, Nong Lam University, Ho Chi Minh City, Vietnam, **2** Department of Computer Science, Faculty of Electrical Engineering and Computer Science, VŠB - Technical University of Ostrava, Ostrava, Czech Republic, **3** Faculty of Information Technology, HUTECH University, Ho Chi Minh City, Vietnam

* nvdu@hcmuaf.edu.vn

## Abstract

This paper proposes a parallel hybrid metaheuristic, named PH-SHOWOA, that integrates the Spotted Hyena Optimizer (SHO) and the Whale Optimization Algorithm (WOA) to solve the Vehicle Routing Problem with Simultaneous Pickup and Delivery and Time Windows (VRPSPDTW). The proposed method leverages the strength of both algorithms: SHO primarily supports population-level diversification, while WOA focuses on best-guided intensification. An adaptive probability control mechanism dynamically regulates the interaction between these two search behaviours during the optimization process. To further enhance robustness and mitigate premature convergence, the framework incorporates simulated-annealing-based acceptance, periodic local search, and population diversification strategies. A parallel implementation enables concurrent solution updates and local refinements, improving computational efficiency on medium-scale instances. The VRPSPDTW is formulated using a hierarchical lexicographic objective that prioritizes minimizing the number of vehicles, followed by total travel distance. Extensive experiments on 65 well-known benchmark instances demonstrate that PH-SHOWOA consistently outperforms standalone SHO and WOA, achieving an average reduction in total distance of over 10%. Compared with advanced algorithms such as Co-GA, MA-FIRD, and ACO-DR, PH-SHOWOA exhibits competitive and often superior performance. Notably, it achieves the lowest total distance on several Rdp and Cdp instances and performs well in centralized-demand scenarios. Furthermore, comprehensive non-parametric statistical tests are conducted to verify the effectiveness and robustness of the proposed method.

## 1 Introduction

In recent years, the rapid growth in consumer demand has significantly accelerated the development of logistics. As a critical component of the supply chain, logistics integrates production, distribution, and delivery processes to ensure goods and

**Data availability statement:** The data underlying the results presented in the study are available from https://github.com/senshineL/VRPenstein/tree/master.

**Funding:** This work was supported in part by the Czech Republic Ministry of Education, Youth, and Sports in the project ROBOPROX under Grant CZ.02.01.01/00/22008/0004590 (to VS), and in part by the European Union under the REFRESH project under Grant CZ.10.03.01/00/22_/0000048 through the Operational Programme Just Transition (to VS).

**Competing interests:** There are no conflicts of interest to declare.

services reach end customers efficiently. Effective vehicle scheduling is pivotal in this process, as it reduces fuel consumption, vehicle maintenance, and labour costs, thereby improving operational efficiency and alleviating financial burdens for companies [1,2].

In the context of globalization and rapid economic growth, logistics has become increasingly important while simultaneously posing significant environmental challenges. Sustainable logistics contributes to mitigating air pollution, global warming, and resource depletion while enhancing economic efficiency by reducing energy and material costs [3,4]. Moreover, this shift reflects a growing change in consumer perception, as more individuals prioritize products and services offered by environmentally responsible companies. Accordingly, efficient vehicle routing is considered a cornerstone of green logistics, since route optimization directly reduces fuel consumption and emissions while enhancing overall service efficiency [3,5,6].

Researchers have widely acknowledged the Vehicle Routing Problem (VRP) as a core topic in Operations Research, especially with respect to sustainable logistics. To capture real-world constraints, many variants have emerged, including VRPPD, VRPTW, and PDPTW, with recent methods [7–10] achieving optimal results. Among these, the Vehicle Routing Problem with Simultaneous Pickup and Delivery and Time Windows (VRPSPDTW) is particularly challenging. As NP-hard [11], it requires vehicles to deliver and collect goods on the same route within strict time windows, which complicates large-scale route planning.

Given the combinatorial nature of VRPSPDTW, exact optimization methods remain tractable only for small instances [12], limiting their applicability in real-world contexts. Consequently, researchers have increasingly turned to heuristics and meta-heuristics, which can produce high-quality solutions within reasonable computational times. The research community has successfully applied classical meta-heuristics — including Genetic Algorithms [8], Tabu Search [13], and Ant Colony Optimization [14] to related VRP variants. However, these approaches often struggle to balance global exploration with local exploitation, leading to premature convergence or stagnation in local optima when applied to VRPSPDTW. Moreover, the growing complexity of logistics systems—including dynamic demand, larger problem sizes, and tighter service constraints—further challenges the scalability and robustness of existing methods.

These limitations highlight the need for advanced hybrid [5,10,15–17] and parallel meta-heuristic approaches [18,19] to explore the solution space while exploiting local improvements effectively. Recent research results have indicated that combining complementary search strategies, incorporating adaptive mechanisms, and leveraging parallel computing can substantially improve solution quality and computational efficiency. Therefore, this study proposes a parallel hybrid metaheuristic combining the Spotted Hyena Optimizer (SHO) and Whale Optimization Algorithm (WOA) with adaptive probability control, periodic local search, and diversification strategies. A simulated-annealing-inspired acceptance criterion is further incorporated to probabilistically accept inferior solutions, enhancing the ability to escape local optima. Parallelization accelerates solution updates and local improvement, ensuring scalability for

large problem instances. This integrated design effectively balances exploration and exploitation, leading to robust performance across multiple benchmark instances. In particular, the proposed method demonstrates strong practical potential for solving VRPSPDTW, thereby improving transportation efficiency and logistics management. The paper proposed an approach to enhance algorithmic performance by accelerating convergence speed and improving the adaptability of the solution process.

The main contributions of this paper are summarized as follows:

1. A parallel hybrid metaheuristic for VRPSPDTW is proposed by integrating the Spotted Hyena Optimizer (SHO) and the Whale Optimization Algorithm (WOA), with clearly separated and complementary roles. SHO promotes population-guided diversification, while WOA performs best-guided intensification. The resulting PH-SHOWOA introduces dedicated information-exchange mechanisms that enable practical cooperation between the two optimization strategies.

2. Efficient neighbourhood structures and a combined local search strategy are designed to jointly reduce the number of vehicles and total travel distance, while strictly preserving feasibility with respect to vehicle capacity and time-window constraints.

3. A parallel evolutionary implementation is developed, in which all individuals evolve concurrently, resulting in reduced computational time and improved scalability on medium-scale VRPSPDTW instances.

4. Extensive computational experiments on 65 public benchmark instances demonstrate that PH-SHOWOA consistently outperforms standalone SHO and WOA, achieving competitive or superior performance compared to several state-of-the-art algorithms in terms of solution quality and robustness.

5. A comprehensive evaluation and validation framework is established, including an ablation study to quantify the contribution of key algorithmic components and a rigorous statistical testing protocol based on Friedman tests, Wilcoxon signed-rank tests, and post-hoc analyses to ensure fair and reliable algorithm comparisons across multiple problem categories.

The remainder of the paper is organized as follows. The Related Works section introduces the problem definition and relevant formulations. The Problem Definition section presents a concise review of related research. The Proposed parallel meta-heuristic for VRPSPDTW section details the proposed parallel hybrid metaheuristic, including its adaptive control mechanisms, local search strategies, and diversification operators. The Experiments and analysis section presents the experimental setup, benchmark datasets, results, and comparative analysis with baseline and advanced algorithms. Some remarks about the findings are discussed in the Discussion section. Finally, the Conclusion and future work section concludes the study, highlighting the key findings and outlining promising directions for future research.

## 2 Related works

The Vehicle Routing Problem (VRP) is a core combinatorial optimization problem in logistics, aiming to determine optimal vehicle routes that serve a set of customers while minimizing total operational cost under various constraints. The classical VRP and most of its variants are NP-hard, and the computational complexity increases rapidly with problem size, making exact optimization methods impractical for real-world applications.

To better capture realistic logistics requirements, the works proposed numerous VRP variants, including the Capacitated Vehicle Routing Problem (CVRP), the Vehicle Routing Problem with Time Windows (VRPTW), and the Vehicle Routing Problem with Pickup and Delivery (VRPPD). Among these, the Vehicle Routing Problem with Simultaneous Pickup and Delivery and Time Windows (VRPSPDTW) is particularly challenging, as it simultaneously considers capacity constraints, bidirectional flows, and strict service time windows. The integration of these constraints significantly enlarges the search space and increases the complexity of the solution.

Early solution approaches for VRPs mainly relied on classical heuristics and local search techniques, such as savings algorithms, tabu search, and variable neighbourhood search. Although effective for small or moderately constrained instances, these methods often suffer from limited scalability and premature convergence when applied to complex VRP variants such as VRPSPDTW. Consequently, metaheuristic algorithms inspired by natural and swarm intelligence have gained increasing attention due to their flexibility and strong global search capability. Swarm intelligence–based algorithms, including genetic algorithms, ant colony optimization, the whale optimization algorithm (WOA), and the spotted hyena optimizer (SHO), have shown promising results for various VRP variants. In particular, the researchers successfully applied SHO-based hybrid frameworks to routing problems. Utama et al. [20] proposed a Hybrid Spotted Hyena Optimizer (HSHO) for the Fuel Consumption Capacitated Vehicle Routing Problem (FCCVRP), explicitly modelling load-dependent fuel consumption and achieving superior performance compared with state-of-the-art algorithms. These results demonstrate that SHO is well-suited for complex VRP variants involving additional operational constraints. Similarly, WOA has proven effective in routing optimization when embedded within hybrid frameworks. Pham et al. [21] introduced a hybrid WOA-based algorithm for the CVRP, reporting improved solution quality, robustness, and scalability compared to existing methods. That confirms that WOA possesses strong global exploration capability for routing problems, particularly when combined with complementary strategies. Despite their effectiveness, single metaheuristic algorithms typically emphasize either exploration or exploitation, which can limit performance on highly constrained VRP variants. As a result, recent research has increasingly focused on hybrid swarm intelligence approaches that integrate complementary search behaviours. In this direction, Abidi et al. [22] proposed a hybrid Spotted Hyena–based Whale Optimization Algorithm (SH-WOA) for automated data classification. Experimental results have demonstrated that the combination of SHO and WOA can significantly improve optimization performance. Although not applied to routing problems, this work provides clear evidence of the feasibility and effectiveness of SHO–WOA hybridization.

To provide a concise overview of existing solution approaches for VRPSPDTW, Table 1 summarizes representative algorithms proposed between 2012 and 2025, covering evolutionary, neighbourhood-based, and hybrid metaheuristic frameworks. Early studies mainly employed genetic algorithms and simulated annealing with hierarchical objective formulations. Later works introduced advanced neighbourhood search, tabu search, memetic algorithms, and ant colony optimization to enhance solution quality and scalability. More recent studies have explored sophisticated hybrid and multi-stage evolutionary strategies, reflecting the increasing complexity of VRPSPDTW.

**Table 1. Summary of the algorithms for VRPSPDTW in the Related works in 2012-2025.**

| Literature | Year | Solution | Objective |
|---|---|---|---|
| Wang et al. [23] | 2012 | co-evolution genetic algorithm (co-GA) | Hierarchical objective |
| Wang et al. [24] | 2015 | parallel simulated annealing (parallel SA) | Hierarchical objective |
| Hof et al. [25] | 2019 | Adaptive large neighborhood search, Path-relinking | Hierarchical objective |
| Shi et al. [26] | 2020 | Variable neighborhood search, tabu search | Hierarchical objective |
| Tang et al. [27] | 2021 | Co-evolution of parameterized search | Weighted total cost |
| Liu et al. [11] | 2021 | Memetic search (MA) | Hierarchical objective |
| Wu et al. [10] | 2023 | Ant colony optimization algorithm, Local Search | Weighted total cost |
| Praxedes et al. [28] | 2024 | (ACO-DR) | Travel cost |
| Lei et al. [29] | 2024 | Branch-Cut-and-Price | Hierarchical objective |
| Zhang et al. [17] | 2025 | Memetic algorithm with feasible and infeasible route descent search (MA-FIRD) Multi-stage hybrid evolutionary multi-objective optimization with a multi-region sampling strategy (MS-HEMO-MRSS) | Hierarchical objective |

In logistics systems, VRPSPDTW extends the classical VRP by jointly considering economic cost, service quality, and environmental impact as competing objectives. Efficiently solving VRPSPDTW, therefore, requires algorithms that can converge toward high-quality trade-off solutions while preserving population diversity. However, traditional exact methods and classical heuristics often suffer from poor scalability and limited ability to achieve an effective exploration–exploitation balance when faced with such complex, multi-constrained search spaces. These limitations have motivated the increasing adoption of advanced metaheuristic and hybrid optimization approaches for VRPSPDTW [12,25,26,28]. While exact optimization techniques become impractical for large-scale problem instances, metaheuristics have demonstrated significant flexibility and adaptability. For example, Parallel Simulated Annealing (p-SA) has been successfully employed in VRPSP-DTW, delivering competitive solutions on benchmark datasets with up to 1000 customers by reducing both travel costs and fleet size [24]. A multi–Markov-chain SA model has further demonstrated improved robustness and solution diversity in pickup-and-delivery contexts [30]. These results emphasize that parallelization is not merely a computational acceleration tool, but also an enabler of enhanced exploration and adaptability in large-scale routing contexts.

Furthermore, the VRPSPDTW has emerged as one of the most difficult VRP extensions due to simultaneous service requirements and additional window constraints. Angelelli and Mansini [12] provided one of the earliest formal formulations, establishing the foundation for subsequent studies. Since then, optimization models have been applied in diverse domains such as home health care scheduling [13], last-mile delivery [31,32], and reverse logistics under uncertain demand [1]. These applications indicate that solving VRPSPDTW is essential for jointly optimizing efficiency, customer satisfaction, and sustainability.

Given the NP-hard nature of VRPSPDTW, exact approaches become impractical for large-scale problems, leading to the adoption of heuristics and metaheuristics. Early contributions explored tabu search, genetic algorithms, and ant colony optimization [8,10,14,33]. Liu et al. [11] introduced memetic search techniques to accelerate convergence, while Lei and Hao [29] and Gibbons and Ombuki-Berman [34] demonstrated the scalability of memetic algorithms for large-scale VRP-SPDTW. Hybrid approaches have also emerged, including product classification–aware models [15], clustering-based heuristics [35], and swarm intelligence enhanced with local search [10]. These studies highlight the effectiveness of combining complementary strategies to achieve a balance between exploration and exploitation.

In parallel, another research stream has focused on uncertainty, robustness, and sustainability. Accordingly, Hu [36] studied VRPs under uncertain demand, while Duan et al. [37] developed robust multi-objective formulations. Fakhrzad et al. [1] proposed a green VRPSPDTW model to minimize fuel consumption, and Elgharably et al. [3] incorporated environmental and customer satisfaction criteria into multi-objective frameworks. Similarly, Yin [4] and Gülmez et al. [6] introduced low-carbon and flexible time-window variants, while Fernández et al. [5] provided a review of heuristics for sustainable routing. Collectively, these works signal a paradigm shift from cost-oriented optimization toward environmentally conscious and resilient routing strategies.

Recent advances in high-performance computing and AI have further shaped VRPSPDTW's variants research. Chen et al. [19] proposed a swarm-parallel framework, and Yao and Chong [18] developed a coevolutionary portfolio approach to improve scalability. Dorneles et al. [38] exploited GPU computing to accelerate metaheuristic experiments, underscoring the importance of computational efficiency for large-scale benchmarks. In parallel, researchers have applied reinforcement learning to VRPTW variants [39], while multitask evolutionary optimization demonstrated knowledge transfer in autonomous transport scenarios [40]. Surveys on machine learning for VRPs [41] suggest a growing frontier where data-driven methods complement heuristic and metaheuristic optimization.

Apart from these, specialized expansion to the VRPSPDTW has attracted much research attention in recent years. Jin & Li [15] explored classification-based pickup–delivery constraints. Ousmane et al. [7] applied a genetic algorithm to a pickup-and-delivery problem with time windows for multi-compartment vehicles, minimizing travel distance and cost with a limited fleet. Wang et al. [35] studied two-echelon multicommodity VRPSPD, and Zhang et al. [42] incorporated two-dimensional loading constraints. To address real-world uncertainty, Elgharably et al. [3] and Wang et al. [35] employed

stochastic and fuzzy programming. Advances in tailored metaheuristics include hybrid genetic–ant colony methods [33] and roulette-wheel PSO variants [43]. More recently, Voigt [44] benchmarked adaptive large-neighbourhood search operators for VRPs, guiding hybrid designs. Meanwhile, Zhou et al. [45] applied multi-objective VRP models to cold-chain logistics, highlighting domain-specific requirements such as product freshness and cold storage.

Overall, research on VRPSPDTW and VRPSPDTW's variants demonstrates a clear progression: from early mathematical models to metaheuristics, from hybrid heuristics to robust and sustainable formulations, and more recently toward parallelized, learning-based, and AI-driven approaches. This trajectory reflects the theoretical complexity of VRPSPDTW and its practical importance in modern logistics, sustainable transportation, and autonomous distribution systems.

Taken together, these contributions illustrate two converging research trajectories:

1) Algorithmic innovation through bio-inspired, hybrid, and parallelized metaheuristics.

2) Application-driven customization of multi-objective VRPs for specialized logistics domains.

Despite these advances, persistent challenges remain—particularly in achieving scalable parallelization, robust performance under uncertainty, and well-distributed Pareto-optimal solutions. These open issues motivate further research into integrated frameworks that unify parallel computing with advanced multi-objective metaheuristics.

Bio-inspired metaheuristics have emerged as a prominent research direction for complex combinatorial optimization, particularly in large-scale routing. Among these, the Spotted Hyena Optimizer (SHO) [46] models the cooperative hunting strategy of spotted hyenas, balancing exploration and exploitation through a dynamic prey–predator mechanism. Its ability to diversify the search space while maintaining convergence pressure has been demonstrated across diverse benchmark functions and engineering design tasks. Similarly, the Whale Optimization Algorithm (WOA) [47] emulates the bubble-net hunting behaviour of humpback whales. WOA is documented for its adaptive exploitation phase and low parameter sensitivity, attributes that contribute to rapid convergence on complex, multimodal landscapes. Comprehensive reviews, such as that of [48], highlight the breadth of WOA applications in engineering optimization and report encouraging results for numerous hybrid variants. Nevertheless, the adoption of WOA in the context of vehicle routing problems (VRP) remains relatively limited compared with classical metaheuristics.

The specific routing variant addressed in this study is the Vehicle Routing Problem with Simultaneous Pickup, Delivery, and Time Windows (VRPSPDTW), which generalizes the classical VRP by requiring each vehicle to perform pickup and delivery operations under strict time-window constraints simultaneously. This combination of bidirectional flows and temporal restrictions substantially enlarges the search space, making exact optimization methods impractical for realistic problem sizes. As a result, existing studies have predominantly focused on metaheuristic-based solution strategies.

As summarized in Table 1, representative approaches include Differential Evolution (DE), Genetic Algorithms (GA), Simulated Annealing (SA), Tabu Search (TS), and their hybrid variants, which have shown competitive performance on benchmark instances. However, these methods primarily rely on single or closely related search mechanisms. To the best of our knowledge, no prior study has explored a hybrid Spotted Hyena Optimizer–Whale Optimization Algorithm (SHO–WOA) for VRPSPDTW, revealing a clear research gap that directly motivates the proposed framework.

Hybridization of metaheuristics has emerged as an effective strategy for addressing the limitations identified above. By integrating algorithms with complementary search behaviours, hybrid frameworks can simultaneously enhance global exploration and local exploitation, accelerate convergence, and reduce the risk of premature stagnation—capabilities that are particularly critical for large-scale and highly constrained routing problems. In parallel, recent research trends increasingly emphasize parallel and distributed metaheuristic architectures, which exploit modern computing resources to improve scalability and computational efficiency in evaluating candidate solutions.

Motivated by these developments, this study proposes a Parallel Hybrid Spotted Hyena Optimizer–Whale Optimization Algorithm (PH-SHOWOA), representing the first application of a SHO–WOA hybrid framework to the VRPSPDTW. The proposed approach leverages SHO's competitive exploration–exploitation mechanism in combination with WOA's adaptive

 

local search capability, while parallelization further enhances scalability by distributing fitness evaluations across multiple processors. By aligning with the broader trajectory of hybrid swarm intelligence research, PH-SHOWOA contributes a novel algorithmic design and provides new empirical evidence for effectively addressing the scalability, complexity, and multi-constraint challenges inherent in VRPSPDTW.

## 3. Problem definition

The Vehicle Routing Problem with Simultaneous Pickup and Delivery and Time Windows (VRPSPDTW) aims to determine a set of feasible routes for a fleet of homogeneous, capacitated vehicles stationed at a central depot. Each vehicle must depart from the depot, serve a subset of customers exactly once, perform both delivery and pickup services within predefined time windows, and return to the depot, while minimizing the total operational cost.

The problem is defined on a directed graph $G = (V, E)$, where $V = \{0, 1, \ldots, n\}$ denotes the set of nodes. Node 0 represents the depot, and $N = V \setminus \{0\}$ is the set of customers. In addition, $E = \{(i, j)|i, j \in V, i \neq j\}$ denotes a set of edges, where $dist(i, j)$, and $time(i, j)$ denote the travel distance and travel time from the node $i$ to node $j$, respectively.

Each customer $i \in N$ is characterized by a delivery demand $d_i$, a pickup demand $p_i$, a service time $s_i$, and a hard time window $[e_i, l_i]$ within which service must start. If a vehicle arrives before $e_i$, it must wait until $e_i$; arrivals after $l_i$ are infeasible. For the depot (node 0), $d_0 = p_0 = s_0 = 0$, and $[e_0, l_0]$ denote the earliest departure time and the latest return time, respectively. A fleet of $J$ identical vehicles is available, each with a capacity $Q$, fixed dispatching cost $u_1$, and unit travel cost $u_2$.

A solution $S$ consists of $K$ routes,

$$S = \{R_1, \ldots, R_K\}, \quad K \leq J \tag{1}$$

where each route $R_k = (h_{k,1}, \ldots h_{k,L_k})$ starts and ends at the depot, i.e., $h_{k,1} = h_{k,L_k} = 0$. The total travel distance of a route $R_k$ is

$$TD(R_k) = \sum_{j=1}^{L_k-1} dist(h_{k,j}, h_{k,j+1}) \tag{2}$$

Arrival and departure times along a route are computed recursively as:

$$arr(h_{k,j}) = dep(h_{k,j-1}) + time(h_{k,j-1}, h_{k,j}), \quad j > 1 \tag{3}$$

$$dep(h_{k,j}) = max\{arr(h_{k,j}), e_{h_{k,j}}\} + s_{h_{k,j}}, \quad j > 1 \tag{4}$$

Vehicle load evolution along a route follows:

$$load(h_{k,j}) = load(h_{k,j-1}) - d_{h_{k,j-1}} + p_{h_{jk,-1}}, j > 1 \tag{5}$$

and must satisfy $load(h_{k,j}) \leq Q$ for all $k$ and $j$.

**Objective Function:** The objective is to minimize the total cost of the solution $S$, denoted by $TC(S)$, which consists of the dispatching cost of the used vehicles and the transportation cost:

$$min\ TC(S) = u_1 K + u_2 \sum_{k=1}^{K} TD(R_k) \tag{6}$$

---

The cost coefficients are chosen such that $u_1 \gg u_2$, thereby enforcing a strict priority on minimizing the number of vehicles $K$ over minimizing the total travel distance. As a result, Eq. (6) effectively implements a lexicographic objective, in which any solution using fewer vehicles is always preferred, and distance minimization is performed only among solutions with the same number of vehicles.

In this study, we focus on a single-objective VRPSPDTW with a hierarchical cost structure. Service quality and environmental considerations are implicitly reflected through time-window feasibility and distance minimization, rather than being modeled as explicit objectives.

This optimization is subject to the following constraints:

1) Fleet size constraint: $K \leq J$

2) Depot constraint: each route starts and ends at the depot,

$$h_{k,1} = h_{k,L_k} = 0, \ \forall k \tag{7}$$

3) Customer coverage constraint: each customer is visited exactly once,

$$\sum_{k=1}^{K} \sum_{j=2}^{L_k-1} 1\left(h_{k,j} = i\right) = 1, \ \forall i \in N \tag{8}$$

4) Capacity constraint:

$$load\left(h_{k,j}\right) \leq Q, \ \forall k,j \tag{9}$$

5) Time-window constraints:

$$dep\left(e_{h_{k,j}}\right) \geq arr\left(h_{k,j}\right) \ \leq l_{h_{k,j}} \ \forall k,j \tag{10}$$

with depot bounds:

$$dep\left(h_{k,1}\right) \geq e_0 \ and \ arr\left(h_{k,L_k}\right) \ \leq l_0 \tag{11}$$

**Remarks**,

- Setting $u_1 = 0$ reduces the problem to minimizing total travel distance only.

- A large ratio $u_1/u_2$ prioritizes minimizing the number of vehicles over travel distance.

The VRPSPDTW is particularly challenging due to the interaction between simultaneous pickup–delivery operations and hard time windows, which significantly enlarges the feasible search space. These characteristics motivate the use of advanced hybrid metaheuristic approaches capable of balancing global exploration and local exploitation to effectively identify high-quality feasible routes.

## 4. Proposed parallel meta-heuristic for VRPSPDTW

In this section, we present the general framework of the Parallel Hybrid based on Spotted Hyena Optimizer & Whale Optimization Algorithm (PH-SHOWOA), developed for the Vehicle Routing Problem with Simultaneous Pickup and Delivery and Time Windows (VRPSPDTW). The flow chart of the algorithm is shown in Fig 1. By combining population-based global search, hybrid SHO–WOA updates, crossover-guided recombination, local search, simulated-annealing

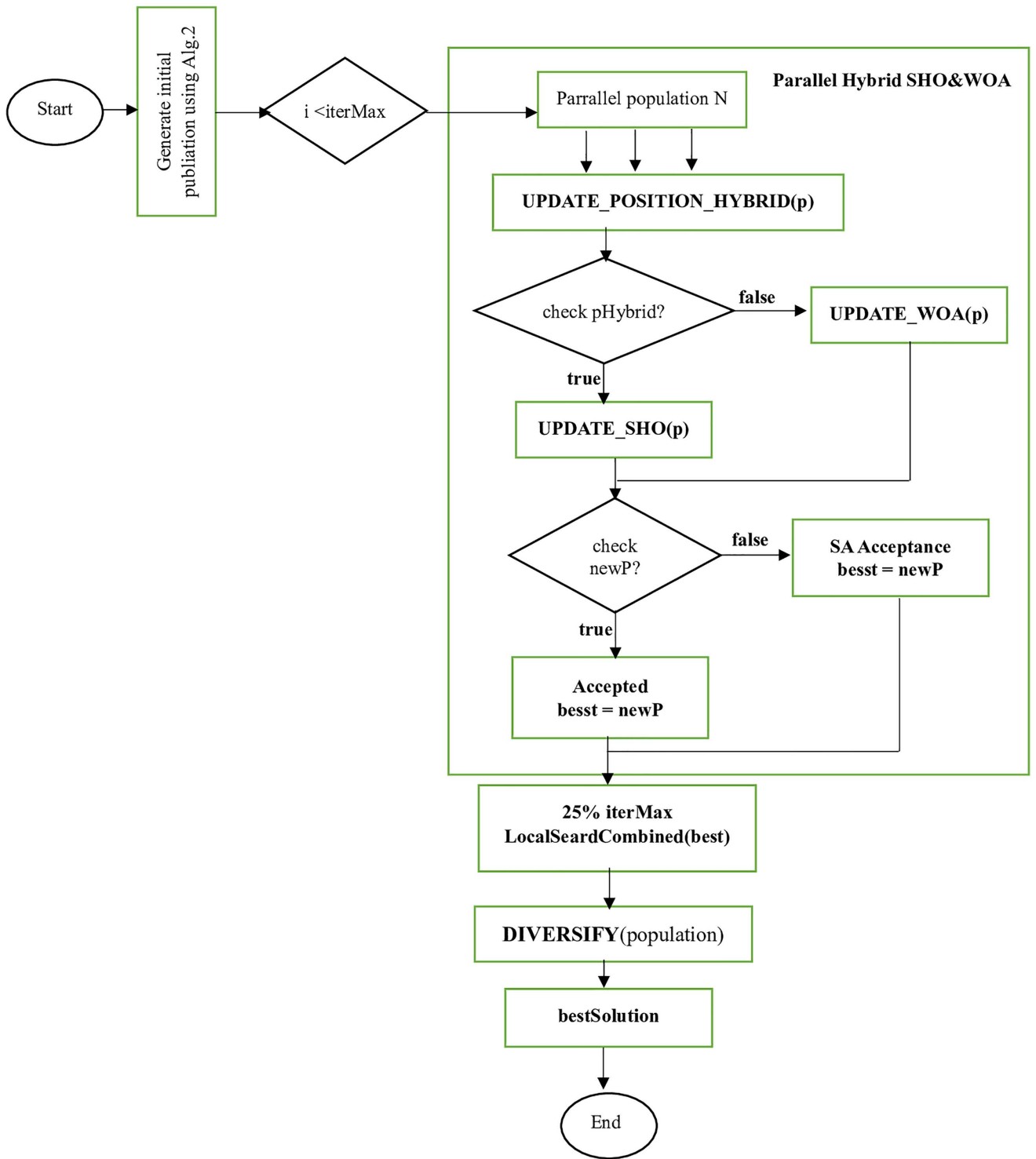

**Fig 1. The flow chart of the proposed Parallel Hybrid based on Spotted Hyena Optimizer & Whale Optimization Algorithm (PH-SHOWOA).**

acceptance, and diversification strategies, the framework ensures robust global exploration, rapid convergence, and resilience against premature local trapping in large-scale VRPSPDTW instances.

## 4.1. General framework of PH-SHOWOA

The study proposes a Parallel Hybrid Spotted Hyena–Whale Optimization Algorithm (PH-SHOWOA) for solving the VRP-SPDTW, as presented in Algorithm 1 (https://github.com/phuongtram/PH-SHOWOA). This algorithm integrates several complementary mechanisms to achieve a well-balanced exploration–exploitation trade-off. Each iteration consists of the following main stages:

- Hybrid SHO–WOA update (Lines 7–11), at each iteration, the control parameters $a$ and $p_{Hybrid}$ are updated (Lines 7–8). Each individual is then evolved using either a Spotted Hyena Optimizer (SHO) operator for global exploration or a Whale Optimization Algorithm (WOA) operator for local exploitation via the UPDATE_POSITION_HYBRID procedure (Lines 10–11), adaptively steering the population toward promising regions of the search space.

- Parallel execution (Lines 10–33), all individuals ("whales") are updated concurrently within a parallel loop (Lines 10–32), and the population is synchronously replaced afterward (Line 33), enabling efficient utilization of multi-core architectures and significantly accelerating convergence on large instances.

- Simulated-annealing acceptance (Lines 17–26): for non-improving candidate solutions, a simulated-annealing-based acceptance criterion is applied (Lines 19–26). Inferior moves are accepted with probability $exp\left(\frac{-\Delta}{T|cf|}\right)$ where the temperature $T$ decreases over iterations (Line 20), helping the search escape local optima.

- Local intensification (Lines 35–37): every fixed number of iterations, a combined local search procedure is applied to the current global best solution (Lines 35–36), intensifying the search around high-quality regions and refining vehicle routes.

- Population diversification (Lines 39–47): when no improvement is detected over a predefined number of iterations (Lines 39–44), a diversification strategy is triggered (Lines 45–46) to perturb the population and prevent premature convergence.

Together, these coordinated mechanisms enable compelling global exploration, rapid convergence, and strong robustness against premature local trapping when solving large-scale VRPSPDTW instances.

### Algorithm 1. Parallel Hybrid SHOWOA with SA Acceptance for VRPSPDTW.

```
Input: initialSolutions, dataProblem, fitness evaluator, feasibility check.
Output: the best solution.
1:  Begin function
2:    Initialize population from initialSolutions
3:    best     ← best individual in population
4:    bestSoFar ← best.fitness
5:    noImprove ← 0
6:    for iteration = 0 to MAX_ITER – 1 do
7:      a ← 2 – 2 * iteration / MAX_ITER      // WOA control parameter
8:      p_Hybrid ← max(0.15, 0.5 * (1 – iteration / MAX_ITER))  // SHO proportion
9:      // --- Parallel hybrid update---
10:     in parallel for each whale w in population do
11:       newSol ← UPDATE_POSITION_HYBRID(w, p_Hybrid, …)
12:       if newSol is infeasible then
13:         return w
14:       end if
```

```
15:         nf ← fitness.CalculateObjective(newSol, D)
16:         cf ← w.fitness
17:         if nf < cf then
18:           w' ← newSol
19:         else    // Simulated-annealing acceptance
20:           T ← 1 - iteration / MAX_ITER
21:           prob ← exp^(-(nf - cf) / (1e-6 + T * |cf|))
22:           if rand() < prob then
23:             w' ← newSol
24:           else
25:             w' ← w
26:           end if
27:         end if
28:         if w'.fitness < best.fitness then
29:           best ← copy(w')
30:           end if
31:           return w'
32:         end parallel
33:         population ← all returned whales
34:         // --- Periodic local search on global best ---
35:         if iteration mod 25 = 0 then
36:             best ← LOCAL_SEARCH_COMBINED(best)
37:         end if
38:         // --- Diversification ---
39:         if best.fitness < bestSoFar - ε then
40:           bestSoFar ← best.fitness
41:           noImprove ← 0
42:         else
43:           noImprove ← noImprove + 1
44:           if noImprove ≥ 50 then
45:             DIVERSIFY(population)
46:             noImprove ← 0
47:           end if
48:         end if
49:     end for
50:   return best
51: End function
```

## 4.2. Initialization

The proposed Algorithm 2 generates a diverse initial solution for VRPSPDTW, building an initial population that supports thorough search space exploration. Its impact on the PH-SHOWOA is evaluated in Experiments and analysis section. The algorithm runs standard Simulated Annealing on route sets, using total travel distance as the objective. It forms neighbourhoods by combining single-route operators (swap, insert, reverse) with inter-route pickup/delivery moves (Pd-Shift, Pd-Exchange). It keeps the current solution whenever a candidate fails capacity or time-window checks. It accepts worse solutions with Boltzmann probability $exp\left(\frac{-\Delta}{T|cf|}\right)$ and reduces the temperature multiplicatively by a constant factor α until it reaches a minimum $T_{min}$. Throughout the search, it records and returns the best feasible solution found.

### Algorithm 2. The Initialization Procedure for VRPSPDTW.

**Input:** solution $S_0$, dataProblem, $T_0$ (initial temperature), α (cooling rate), $T_{min}$, $iter_{max}$, fitness evaluator, feasibility check.
**Output:** an improved solution S*.
1: **Begin function**

```
2:    S ← S₀; S* ← S; E* ← fitness.CalculateObjective(S); T ← T₀
3:    While T > T_min do
4:      For i = 1 to iter_max do
5:        S' ← PERTURB(S) // apply local move (swap, insert, reverse, Pd-shift, Pd-exchange)
6:        If IS_FEASIBLE(S') then// check capacity and time windows
7:          Δ ← fitness.CalculateObjective (S') – fitness.CalculateObjective (S)
8:          If Δ < 0 or rand(0,1) < exp^(-Δ / T|cf|) then
9:            S ← S'
10:             If totalCost(S) < E* then
11:               S* ← S; E* ← fitness.CalculateObjective (S)
12:           End If
13:         End If
14:       End If
15:     End For
16:     T ← α × T
17:   End While
18:   Return S*
19:   End function
```

## 4.3. Strategy for a parallel hybrid based on SHO–WOA framework

The proposed framework employs a hybrid individual-update mechanism that leverages the complementary strengths of the Spotted Hyena Optimizer (SHO) and the Whale Optimization Algorithm (WOA), as outlined in Algorithm 3. At each iteration, every solution candidate ("whale") is updated independently and in parallel via the **UPDATE_POSITION_HYBRID** procedure, enabling scalable search on large VRPSPDTW instances.

The exploration–exploitation balance is explicitly controlled by the probability parameter $p_{Hybrid}$. For each update, a random number $r \in [0, 1]$ is generated (line 2 of Algorithm 3), and one of the following update strategies is selected:

- If $r < p_{Hybrid}$, the candidate follows the **BasedOnSHOUpdate** operator **(Lines 3–6)**, which emphasizes global exploration. A tournament selection mechanism is first applied to identify a strong partner solution (Line 5). A guided crossover combining information from the global best, the selected partner, and the current solution is then executed, optionally followed by mutation operators such as swap, 2-opt, or relocate. Feasibility checks and automatic repair ensure that vehicle-capacity and time-window constraints are preserved. This update mode promotes broad exploration and mitigates premature convergence.

- Otherwise, the candidate is refined using the **BasedOnWOAUpdate** operator (**Lines 7–9**), which focuses on local exploitation. This operator follows the classical WOA mechanisms, where adaptive coefficient vectors $A$ and $C$ govern encircling, spiral movement, and randomized route-level operations. These mechanisms intensify the search for high-quality solutions while maintaining feasibility.

By dynamically alternating between these two update modes within a parallel execution environment, the proposed framework achieves:

- Adaptive exploration–exploitation balance, such as SHO expands the search space, whereas WOA refines promising regions.

- Population diversity and robustness guided crossover, mutation, and feasibility repair to reduce the risk of stagnation.

- Accelerated convergence: independent parallel updates across all individuals effectively exploit multi-core hardware, reducing computational time for large-scale VRPSPDTW instances.

Overall, this hybrid update strategy forms the core of PH-SHOWOA, providing a flexible and effective mechanism for navigating the complex and constrained search space of the VRPSPDTW.

**Algorithm 3. UPDATE_POSITION_HYBRID (Exploration–Exploitation Control).**

```
Input: population, t, T_max, p_Hybrid, dataProblem, fitness evaluator, feasibility check.
Output: the best candidate
1:  Begin function
2:    r ← UniformRandom(0, 1)
3:    if r < p_Hybrid then
4:      // SHO-based diversification (exploration)
5:      partner ← TournamentSelect(population, k = 3)
6:      candidate ← BasedOnSHOUpdate(current, partner)
7:    else
8:      // WOA-based intensification (exploitation)
9:      candidate ← BasedOnWOAUpdate(current)
10:   end if
11:    return candidate
12: End function
```

In this hybrid framework, the SHO operator is intentionally used for population-guided diversification (Based on the SHO update operator section). In contrast, the WOA operator is reserved for best-guided intensification (Based on the WOA update operator section), ensuring a clear and consistent separation between exploration and exploitation.

## 4.4. Based on the WOA update operator

Therefore, the SHO and WOA components play complementary yet non-overlapping roles: SHO expands the search space through exploration and diversification (Strategy for a Parallel Hybrid Based on SHO–WOA Framework section), whereas WOA exclusively refines high-quality regions via best-guided local intensification (Based on the WOA update operator section), eliminating any ambiguity in their functional responsibilities.

The WOA is employed exclusively as a local intensification operator within the proposed hybrid framework. Unlike the SHO component, which is responsible for global exploration and population diversification, the WOA update is strictly best-guided and focuses on refining promising regions of the search space.

As summarized in Algorithm 4, each candidate solution ("whale") updates its position with respect to the current global best solution by mimicking the cooperative hunting behaviour of humpback whales. Specifically, the update alternates among encircling the best solution, spiral movement toward the best solution, and controlled local route perturbations triggered only when the encircling condition is not satisfied. Importantly, all these mechanisms are anchored to the current best solution, ensuring exploitation rather than global exploration.

Feasibility checks and repair mechanisms are applied immediately after each route-level or multi-route modification (Lines 25–27 and 34–36), guaranteeing that vehicle capacity and time-window constraints are preserved. In rare cases where global feasibility cannot be restored, a fallback reconstruction strategy is applied (Lines 38–40) to maintain algorithmic robustness.

Through this strictly best-guided update design, the WOA operator intensifies the search around high-quality solutions without introducing uncontrolled randomness, thereby complementing the exploration-oriented SHO component and ensuring a clear and consistent exploration–exploitation separation across the hybrid framework.

**Algorithm 4. BasedOnWOAUpdate (Best-Guided Intensification).**

```
Input: whale, bestWhale, dataProblem, fitness evaluator, feasibility check.
Output: updated bestWhale
1:Begin function
2:    currentSol ← whale.getSolution
3:    bestSol   ← bestWhale.getSolution
4:    newSol    ← deep copy of currentSol
```

```
5:    newRoutes  ← newSol.getRoutes
6:    dim     ← number of routes in newRoutes
7:    bestDim    ← number of routes in bestSol
8: limit     ← min(dim, bestDim)
9: A ← RandomVector(dim, a, true)  // A ∈ [-a, a]
10: C ← RandomVector(dim, 2.0, false)  // C ∈ [0, 2]
11: for i = 0 … limit-1 do
12:      p ← UniformRandom(0,1)
13:      route ← newRoutes[i]
14:      bestRoute ← bestSol.routes[i]
15:      backupRoute ← copy(route)
16:      if p < 0.5 then
17:        if |A[i]| < 1 then
18:          ENCIRCLING_PREY(route, bestRoute, A[i], C[i])
19:        else
20:          if UniformRandom(0,1) < 0.5 then
21:            ApplyRandomOperation(route) (Li&Lim [49])// swap, insert, 2-opt
22:          else
23:            backupRoutes ← deep copy of newRoutes
24:          ApplyRandomMultiRouteOperation(newRoutes) (Li&Lim [49])//Pd-shift, Pd-exchange
25:            if NOT check.ValidateRoutes(newRoutes, customers, travelTime) then
26:              newRoutes ← backupRoutes  // rollback if infeasible
27:            end if
28:          end if
29:        end if
30:      else
31:        SPIRAL_MOVEMENT(route, bestRoute, C[i])
32:      end if
33:      if NOT check.IsRouteFeasible(route, customers, capacity, travelTime) then
34:        newRoutes[i] ← backupRoute
35:      end if
36:    end for
37:    if NOT check.ValidateRoutes(newRoutes, customers, travelTime) then
38:      newRoutes ← FallbackSingleCustomerRoutes()
39:    end if
40:    newFitness ← fitness.CalculateObjective(newRoutes)
41:    candidate ← Solution(newRoutes, newFitness)
42:    if newFitness < whale.fitness then
43:      whale.solution ← candidate
44:      whale.fitness ← newFitness
45:      if newFitness < bestWhale.fitness then
46:          bestWhale.solution ← candidate.copy()
47:          bestWhale.fitness ← newFitness
48:        end if
49:      end if
50:    return bestWhale
51: End function
```

## 4.5. Based on the SHO update operator

In contrast to the WOA-based operator in Based on the WOA update operator section, which focuses on best-guided intensification, the SHO update operator (Algorithm 5) is explicitly designed as a population-guided diversification mechanism within the hybrid framework. Its primary role is to expand the search space and maintain population diversity by recombining information from multiple high-quality solutions rather than strictly following the current global best.

Specifically, the SHO operator first generates a guided offspring by blending information from the current solution, the global best solution, and a partner selected from the population. This recombination promotes exploration by introducing structural variations that may not be reachable through local refinement alone. A probabilistic mutation, applied with a small fixed probability, further increases diversity and helps the search escape local optima. All offspring solutions are subjected to feasibility checks and repair procedures to ensure compliance with capacity and time-window constraints. Finally, the offspring is evaluated and compared against the reference solution, and only the feasible solution with the lower objective value is retained.

By emphasizing population-driven recombination and controlled mutation, the SHO update operator complements the exploitative behaviour of WOA. This clear functional separation—SHO for diversification and global exploration, WOA for intensification and local exploitation—ensures a consistent and well-defined exploration–exploitation balance in the proposed PH-SHOWOA framework.

### Algorithm 5. BasedOnSHOUpdate (Population-Guided Diversification).

```
Input: hyena, bestHyena, dataProblem, fitness evaluator, feasibility check.
Output: best candidate
1:  Begin function
2:  base ← deep copy of current     // preserve original solution
3:    // Create a guided offspring using best and partner
4:    combined ← CrossoverGuided (bestHyena, partner, current)
5:    // With probability 0.35 apply a local mutation to the offspring
6:    if Random() < 0.35 then
7:      combined ← MutationOperators (combined, fitness, check)
8:    end if
9:    // If offspring is infeasible and cannot be repaired, keep the base solution
10:   if NOT Feasible_Or_Repair(combined) then
11:     return base
12:   end if
13:   // Evaluate objective values (lower is better)
14:   fComb ← fitness.CalculateObjective (combined)
15:   fBase ← fitness.CalculateObjective(base)
16:   // Return the better solution
17:   if fComb < fBase then
18:     return combined
19:   else
20:     return bestHyena
21:   end if
22: End function
```

To avoid excessive selection pressure toward the global best, the offspring generated by the SHO operator is compared against the current solution rather than directly against the global best, ensuring that this operator primarily contributes to diversification rather than local intensification.

### 4.6. Crossover operator

The crossover operator (Algorithm 6) selects one or two high-quality route segments from the best solution (Lines 7–18), merges the remaining customers drawn from both the partner and current parents (Lines 19–29), and reconstructs a complete set of depot-closed routes while enforcing the vehicle-capacity constraint (Lines 35–50). Finally, it evaluates the offspring's objective value using the standard distance-based cost function (Line 53).

This design preserves elite genetic material, promotes population diversity, and guarantees feasibility for the VRPSP-DTW. The guided crossover, therefore, combines elitism, diversity, feasibility, and efficiency in a single operator: it inherits strong route segments, blends customer assignments from multiple parents, and applies capacity checks during construction to consistently generate high-quality, valid offspring for large-scale, multi-objective VRP scenarios.

**Algorithm 6. CrossoverGuided for VRPSPDTW.**

**Input:** solutions best, partner, current, dataProblem, fitness evaluator, feasibility check.
**Output:** offspring –a new feasible solution

```
1:  Begin function
2:    if best.routes is empty then
3:      return copy(current)
4:    end if
5:    offspring ← empty solution
6:    SeedRoutes ← ∅
7:    KeptCustomers ← ∅
8:    // --- Select seed routes from the best solution ---
9:    if |best.routes| = 1 then
10:     take ← 1
11:   else
12:     take ← 1 with probability 0.6, else 2
13:   end if
14:   indices ← random permutation of {1,…,|best.routes|}
15:   for k ← 1 to take do
16:     segment ← best.routes[indices[k]] without depot nodes
17:     SeedRoutes ← SeedRoutes U {segment}
18:     KeptCustomers ← KeptCustomers U segment
19:   end for
20:   // --- Collect remaining customers from partner and current ---
21:   Remaining ← ∅
22:   for parent ∈ {partner, current} do
23:     for route ∈ parent.routes do
24:       for c ∈ route \ {0} do
25:         if c ∉ KeptCustomers then
26:           Remaining ← Remaining U {c}
27:         end if
28:       end for
29:     end for
30:   end for
31:   // --- Initialize offspring with depot-closed seed routes ---
32:   OffspringRoutes ← ∅
33:   for segment ∈ SeedRoutes do
34:     OffspringRoutes ← OffspringRoutes U {[0] + segment + [0]}
35:   end for
36:   // --- Insert remaining customers while respecting capacity ---
37:   CurrentRoute ← [0]
38:   CurrentLoad ← 0
39:   for c ∈ Remaining do
40:     demand ← customers[c].pickup - customers[c].delivery
41:     if |CurrentRoute| > 1 and CurrentLoad + demand > capacity then
42:       CurrentRoute ← CurrentRoute + [0]
43:       OffspringRoutes ← OffspringRoutes U {CurrentRoute}
44:       CurrentRoute ← [0]
45:       CurrentLoad ← 0
46:     end if
47:     CurrentRoute ← CurrentRoute + [c]
48:     CurrentLoad ← CurrentLoad + demand
49:   end for
50:   if |CurrentRoute| ≥ 2 then
51:     CurrentRoute ← CurrentRoute + [0]
52:     OffspringRoutes ← OffspringRoutes U {CurrentRoute}
53:   end if
```

```
54:    // --- Final evaluation ---
55:    offspring.routes ← OffspringRoutes
56:    offspring.fitness ← fitness.CalculateObjective(OffspringRoutes)
57:    return offspring
58: End function
```

## 4.7. Mutation operator

The mutation operator (Algorithm 7) enhances the SHO search by providing local intensification, population diversity, feasibility control, and computational efficiency. It strategically perturbs feasible solutions and instantly discards any infeasible changes, ensuring that every accepted offspring remains high-quality and valid. The operator combines fine-grained intra-route adjustments (SWAP at line 9) with structural inter-route changes (RELOCATE at line 14). Built-in rollback safeguards (line 16) maintain feasibility throughout, achieving a balanced mix of diversification and intensification suited to large-scale VRPSPDTW instances.

It works as follows: the operator randomly selects either **SWAP**, which exchanges two customers within a single route, or **RELOCATE**, which moves a customer from one route to another. After applying the chosen mutation, it checks route feasibility—including capacity and time-window constraints—and restores the backup if any constraint is violated. Finally, it recalculates the objective value and returns the improved solution, or the unchanged backup when no improvement is feasible.

### Algorithm 7. MutationOperators.

```
Input: currentSolution (S), dataProblem, fitness evaluator, feasibility check.
Output: the S' solution
1:  Begin function
2:     S' ← DeepCopy(S)
3:     R ← S'.Routes
4:     if R is empty then
5:        return S'
6:     end if
7:     mutationType ← UniformRandom({SWAP, RELOCATE}) // Choose operator
8:     i ← UniformRandom(0, |R| − 1)
9:     RouteBackup ← Copy(R[i])
10:    if mutationType = SWAP then
11:       ApplyRandomOperation (R[i]) (Li&Lim [49])//apply local move (swap, insert, reverse)
12:       if not ValidateRoutes(R, Customers, TravelTime) then
13:          R[i] ← RouteBackup // Roll back if constraints violated
14:       end if
15:    else if mutationType = RELOCATE and |R| ≥ 2 then
16:       RoutesBackup ← DeepCopy(R) // Full backup for multi-route operation
17:          ApplyRandomMultiRouteOperation(newRoutes) (Li&Lim [49])//Pd-shift, Pd-exchange two routes
18:       if not ValidateRoutes(R, Customers, TravelTime) then
19:          R ← RoutesBackup // Roll back if constraints violated
20:       end if
21:    end if
22:    S'.Fitness ← fitness.CalculateObjective(R) // Recalculate objective
23:    return S'
24: End function
```

## 4.8. Combined Local Search

This combined local search balances diversification (cross-route relocations and swaps) and intensification (2-opt edge improvements) while strictly enforcing feasibility. It efficiently drives solutions toward fewer routes and shorter travel distances, making it an excellent post-processing step for metaheuristics such as the hybrid SHO–WOA.

Algorithm 8 sequentially explores three neighbourhoods—2-opt for intra-route edge reversal, Relocate for single-customer reinsertion, and Swap for inter-route customer exchange—using a first-improvement policy that accepts the first neighbour with a strictly lower cost to speed convergence. Each candidate move is verified or repaired through **feasibleOrRepair**, guaranteeing continuous satisfaction of vehicle-capacity and time-window constraints.

The cost evaluation employs **CALCULATE_OBJECTIVE** with a hierarchical objective: first assigns a significant penalty to the number of routes (to prioritize route minimization) and then minimizes total travel distance among solutions with the exact route count. This layered evaluation preserves the search mechanism while ensuring that route reduction always dominates distance reduction.

Through this integrated design, the combined local search delivers substantial local intensification, maintains feasibility, and naturally supports large-scale VRPSPDTW optimization.

**Algorithm 8. CombinedLocalSearch.**

```
Input: S, dataProblem, fitness evaluator, feasibility check.
Output: improved solution S*
1:  S* ← S; bestCost ← fitness.CalculateObjective (S*)
2:  improved ← true
3:  while improved do
4:    improved ← false
5:    // (a) 2-opt: intra-route edge reversal
6:    for each route r in S* do
7:      for each edge pair (i,j) in r with 1 ≤ i < j−1 do
8:        S' ← reverse segment i..j in r
9:        if feasibleOrRepair(S') and fitness.CalculateObjective (S') < bestCost then
10:           S* ← S'; bestCost ← fitness.CalculateObjective (S'); improved ← true; break
11:        end if
12:      end for
13:    end for
14:    if improved then continue
15:    // (b) Relocate: move one customer to another position/route
16:    for each customer c in S* do
17:      for each feasible insertion position p ≠ current of c do
18:        S' ← relocate c to p
19:        if feasibleOrRepair(S') and fitness.CalculateObjective (S') < bestCost then
20:           S* ← S'; bestCost ← fitness.CalculateObjective (S'); improved ← true; break
21:        end if
22:      end for
23:    end for
24:    if improved then continue
25:    // (c) Swap: exchange customers between routes
26:    for each pair of customers (c1,c2) in distinct routes do
27:      S' ← swap c1 and c2
28:      if feasibleOrRepair(S') and fitness.CalculateObjective (S') < bestCost then
29:         S* ← S'; bestCost ← fitness.CalculateObjective (S'); improved ← true; break
30:      end if
31:    end for
32:  end while
33:  return S*
```

Feasibility control follows a strict two-stage pipeline. Algorithm 9 performs feasibility detection without modifying the solution and immediately rejects any violation. Algorithm 10 is invoked only when infeasibility is detected and applies targeted repair operations, with Algorithm 9 repeatedly re-applied to validate the repaired solution.

## 4.9. Feasible and infeasible routes for VRPSPDTW

This procedure verifies the feasibility of a candidate solution by sequentially scanning all routes. For each route, the vehicle load and arrival time are incrementally updated along the visiting sequence. The algorithm explicitly checks all VRP-SPDTW constraints, including **(i)** vehicle capacity limits with simultaneous pickup and delivery, **(ii)** customer time-window satisfaction, and **(iii)** the exactly-once visitation constraint, ensuring that no customer is duplicated or omitted. The solution is immediately declared infeasible as soon as any constraint violation is detected.

### Algorithm 9. FeasibilityCheck.

```
Input: Solution S, customer data, capacity, travel-time matrix
Output: true if the solution is feasible; false otherwise.
1:  Initialize visited[1..N] ← false
2:  for each route r ∈ S do
3:    load ← 0; time ← 0
4:    for each consecutive customer pair (i,j) in r do
5:      time ← time + travelTime(i,j)
6:      if j ≠ depot then
7:        if time < readyTime(j) then time ← readyTime(j)
8:        if time > dueTime(j) then return false // time-window violation
9:        load ← load + delivery(j) – pickup(j)
10:       if load < 0 or load > capacity then return false // capacity violation
11:        if visited[j] = true then return false // duplicate visit
12:        visited[j] ← true
13:        time ← time + serviceTime(j)
14:      end if
15:    end for
16:  end for
17:  if ∃ customer k with visited[k] = false then return false // missing customer
18:  return true
```

## 4.10. Feasibility–Repair Mechanism for VRPSPDTW

Algorithms 9 and 10 together form a strict feasibility–repair pipeline for VRPSPDTW solutions. Algorithm 9 (Lines 1–18) performs *pure feasibility detection* without altering the solution, sequentially verifying capacity constraints, time-window feasibility, and the exactly-once customer visitation requirement.

Algorithm 10 is invoked only when Algorithm 9 reports infeasibility. It applies targeted repair actions corresponding to the detected violation types, including duplicate or missing customer visits, capacity overloads, and time-window conflicts. After each repair phase, Algorithm 9 is re-applied to revalidate feasibility.

If all violations are successfully resolved, the repaired solution is returned; otherwise, the original solution is preserved, and the repair attempt is declared unsuccessful. This explicit detect–repair–recheck design ensures correctness, transparency, and reproducibility of the feasibility handling process.

### Algorithm 10. RepairProcedure.

```
Input: Infeasible solution S, customer data, capacity, travel-time matrix
Output: Repaired feasible solution S'or failure
1: 1: S' ← deep copy of S
2: // Step 1: Repair duplicate and missing customers (exactly-once constraint)
3: Identify customers visited more than once in S'
4: for each duplicated customer c do
5:   Keep c in the route with the best insertion cost
6:   Remove c from all other routes
7: end for
```

```
8:  Identify customers not visited in S'
9:  for each unvisited customer c do
10:    Insert c into the best feasible position
11:    if insertion fails then
12:      Create a new route containing c
13:    end if
14:  end for
15:  // Step 2: Repair capacity violations
16:  for each route r in S' do
17:    while load(r) > Q or load(r) < 0 do
18:      Select customer c with the largest |delivery(c) - pickup(c)|
19:      Remove c from r
20:      Insert c into another route with feasible capacity
21:      if no feasible route exists then
22:        Open a new route for c
23:      end if
24:    end while
25:  end for
26:  // Step 3: Repair time-window violations
27:  for each route r in S' do
28:    Compute arrival times along r
29:    while exists customer c with arrivalTime(c) > dueTime(c) do
30:      Remove c from r
31:     Reinsert c at the best feasible position (earliest arrival)
32:      if reinsertion fails then
33:        Assign c to a new route
34:      end if
35:    end while
36:  end for
37:  // Step 4: Local route repair
38:  for each route r in S' do
39:    Apply intra-route 2-opt to reduce travel time and delay
40:  end for
41:  // Step 5: Final validation
42:  if FeasibilityCheck(S') = true then // Algorithm 9
43:    return S'
44:  else
45:    return S // repair unsuccessful
46:  end if
```

## 4.11. Computational complexity analysis

Let $N$ denote the number of customers, $K$ the number of vehicles, $P$ the population size, and $I$ the maximum number of iterations. In Algorithm 1, all individuals are updated in parallel at each iteration (Lines 10–32). For a single individual, the dominant computational cost arises from the UPDATE_POSITION_HYBRID procedure, which includes SHO/WOA position updates, feasibility checking (Algorithm 9), and objective evaluation. These operations require at most a full scan of the customer set and therefore have worst-case complexity $\mathcal{O}(N)$.

As a result, the computational complexity per iteration is $\mathcal{O}(P \cdot N)$, and the overall time complexity of PH-SHOWOA is $\mathcal{O}(I \cdot P \cdot N)$. The periodic local search applied exclusively to the global best solution (Lines 35–37) incurs an additional $\mathcal{O}(N)$ cost every fixed number of iterations and does not affect the asymptotic complexity.

The parallel implementation distributes the $P$ individual updates across multiple processing cores, thereby reducing effective runtime while preserving the same theoretical complexity. This design improves scalability on medium- and large-scale VRPSPDTW instances.

## 4.12. Discussion

This section discusses the technical rationale underlying the proposed PH-SHOWOA framework. The hybridization of SHO and WOA is motivated by their complementary search roles. The SHO-based update promotes population-guided diversification through crossover and mutation, facilitating exploration of diverse routing structures. In contrast, the WOA-based update performs best-guided intensification by refining solutions around the current global best. The adaptive probability parameter $p_{Sho}$ enables a gradual transition from exploration to exploitation, ensuring a balanced search process.

A key design decision in PH-SHOWOA is the selective use of computationally expensive local search. Unlike memetic algorithms that apply intensive neighborhood exploration to most offspring at every generation, PH-SHOWOA restricts combined local search to periodic refinement of the global best solution. This strategy significantly reduces the computational cost per iteration while retaining a strong intensification capability where it is most beneficial.

The hierarchical objective structure further guides the search toward practically relevant solutions. By prioritizing vehicle minimization before distance reduction, the algorithm naturally favours compact route sets, while distance optimization is performed only among solutions with the same fleet size. That aligns with operational objectives in real-world logistics and simplifies objective handling compared with fully multi-objective formulations.

Parallelization is implemented at the population-update level, where individuals are evolved independently and synchronously. This coarse-grained parallel design minimizes synchronization overhead and enables efficient utilization of multi-core architectures. As a result, PH-SHOWOA achieves favourable scalability on medium-sized VRPSPDTW instances without introducing excessive algorithmic complexity.

Overall, PH-SHOWOA reflects a deliberate balance between diversification, intensification, and computational efficiency. The combination of lightweight swarm-based updates, adaptive hybrid control, and targeted local refinement provides a robust and scalable framework for solving highly constrained VRPSPDTW instances.

## 5. Experiments and analysis

A well-structured benchmark dataset is essential for evaluating algorithms for the Vehicle Routing Problem with Simultaneous Pickup and Delivery and Time Windows (VRPSPDTW). The selected benchmarks assess performance along two orthogonal dimensions—geographical customer distribution and constraint intensity—allowing a rigorous evaluation of algorithm robustness, efficiency, and practical applicability. The proposed PH-SHOWOA was compared with state-of-the-art methods across all benchmark datasets, as summarized in [23]. We design experiments to test the algorithm's performance under diverse problem characteristics and varying levels of complexity.

### 5.1. Benchmark datasets

The experiments carried out using the benchmark dataset from Wang & Chen (WC) [23], considering both time-window and capacity constraint intensities. Instances in the WC set consist of six subsets—C1, C2, R1, R2, RC1, and RC2—where Cdp indicates clustered customer locations, Rdp indicates uniformly random locations, and RCdp indicates a mix of clustered and random locations. Type 1 instances have narrow time windows and small vehicle capacities, while Type 2 instances have wide and large ones. The proposed algorithms were developed, with Eq. (6) employed to evaluate solution quality as flows: minimizing NV (number of vehicles used) for the WC set is the primary objective, with TD (total distance) as secondary. In Eq. (6), the ratio of dispatching cost per vehicle ($u_1$) to unit travel cost ($u_2$) is set sufficiently large—following Liu et al. [11], $u_1 = 2000$ and $u_2 = 1$—to reflect this priority in Tables 3–6, 10 and 11.

### 5.2. Compared algorithms and settings

In our comparative study of the VRPSPDTW problem, we evaluated three state-of-the-art algorithms: Co-GA [23], MA-FIRD [29], and ACO-DR [14]. Researchers have tested these methods on all or part of the benchmark instances, so we used the best results reported in their original publications as reference values.

To assess the proposed algorithm comprehensively, we organized the experiments into three parts:

1) PH-SHOWOA vs. SHO and WOA on small-scale WC instances.

2) PH-SHOWOA vs. SHO and WOA on medium-scale WC instances.

3) PH-SHOWOA vs. Co-GA, MA-FIRD, and ACO-DR on medium-scale WC instances.

We implemented the original SHO [46], original WOA as WOA [47], and PH-SHOWOA in Java. We ran them 30 times on a laptop equipped with a 12th Gen Intel® Core™ i5-1235U processor (base frequency 1.30 GHz) and 16 GB RAM. Table 2 lists the parameter settings used in our experiments.

**Parameter selection:** the SHO and WOA parameters follow the recommendations of their original studies to ensure fairness and reproducibility. In SHO, the control parameter $h$ is linearly decreased from 5 to 0 to shift from exploration to exploitation gradually. At the same time, the random coefficients $rd_1$ and $rd_2$ are uniformly sampled from [0, 1] at each iteration to preserve diversity. In WOA, the parameter $a$ decreases linearly from 2 to 0, enabling a smooth transition from global exploration to local exploitation, and the random factor $r \in [0, 1]$ controls stochastic movements around the current best solution. For PH-SHOWOA, the hybrid control parameter $p_{Hybrid}$ is adaptively adjusted during the search (Algorithm 1, Lines 7–9), progressively reducing the proportion of SHO-based updates and increasing WOA-based updates as iterations proceed. This time-varying schedule mitigates sensitivity to manual tuning and enforces the intended exploration–exploitation trade-off.

Sensitivity considerations: a limited sensitivity analysis on small and medium size WC instances indicates that the proposed algorithm is relatively robust to moderate variations in $p_{Hybrid}$, $h$, and $a$. Based on these observations, the parameter settings reported in Table 2 are used for all experiments.

For Co-GA, MA-FIRD, and ACO-DR, we adopted the results directly from their original papers. The detailed comparative results and analysis appear below.

**5.2.1. Comparison with SHO, WOA on small-scale WC instances.** This section compares the proposed algorithm PH-SHOWOA with the original algorithms SHO and WOA on 9 small WC instances. All three methods solved these problems quickly and consistently, showing only minor differences in computation time. Because these cases involve narrow time windows and small vehicle capacities, they pose little challenge to any algorithms. Table 3 reports the detailed results. The first column lists the instance name, and the second column gives the instance size. For each algorithm, we show the values of NV, TD, and runtime t (ms). The last columns present the gaps between PH-SHOWOA and the other two algorithms. Where

$$GAP_{NV} = \{NV_{PH-SHOWOA} - NV_{SHO} \quad NV_{PH-SHOWOA} - NV_{WOA}$$

$$GAP_{TD} = \{\frac{(TD_{PH-SHOWOA} - TD_{SHO})}{TD_{SHO}} x100\% \quad \frac{(TD_{PH-SHOWOA} - TD_{WOA})}{TD_{WOA}} x100\%$$

**Table 2. The parameter settings.**

| Parameter | Value/ Range | Notes |
|---|---|---|
| Number of iterations | 1000 | Fixed across all algorithms |
| Initial population | 30 | Fixed across all algorithms |
| Number of runs | 30 | Independent runs |
| $p_{Hybrid}$ (PH-SHOWOA) | (0, 1) | Weight between SHO and WOA components |
| h (SHO) | [5 ◊0] | Linearly decreases to balance exploration vs. exploitation |
| the $rd_1$, $rd_2$ (SHO) | [0, 1] | Random factors refreshed each iteration for diversity |
| a (WOA) | [2 ◊ 0] | Controls exploration–exploitation transition |
| r (WOA) | [0,1] | Random number enabling agents to move diversely around the current best solution |

**Table 3. Comparison of experimental results with SHO, WOA on small WC instances.**

| Instance | Size | SHO | | | WOA | | | PH-SHOWOA | | | GAP | |
|---|---|---|---|---|---|---|---|---|---|---|---|---|
| | | NV | TD | t(ms) | NV | TD | t(ms) | NV | TD | t(ms) | $NV_{SHO/WOA}$ | $TD_{SHO/WOA}$(%) |
| RCdp1001 | 10 | 2 | 323.74 | 24.42 | 3 | 365.25 | 24.65 | 3 | 369.74 | 97.30 | 1/ 0 | 14.21/ 1.23 |
| RCdp1004 | 10 | 2 | 171.01 | 17.50 | 2 | 216.69 | 14.12 | 2 | 172.48 | 79.42 | 0/ 0 | 0.86/ **−20.40** |
| RCdp1007 | 10 | 3 | 224.49 | 22.46 | 2 | 246.27 | 16.30 | 2 | 234.22 | 102.70 | **−1**/ 0 | 4.33/ **−4.89** |
| RCdp2501 | 25 | 5 | 652.16 | 40.22 | 6 | 655.73 | 37.87 | 6 | 648.15 | 268.01 | 1/ 0 | **−0.61/ −1.16** |
| RCdp2504 | 25 | 5 | 517.79 | 50.63 | 4 | 537.45 | 43.81 | 5 | 499.36 | 368.53 | 0/ 1 | **−3.56**/ 38.09 |
| RCdp2507 | 25 | 5 | 619.33 | 59.30 | 5 | 589.65 | 48.56 | 5 | 536.28 | 383.19 | 0/ 0 | **−13.41/ −9.05** |
| RCdp5001 | 50 | 11 | 1364.51 | 134.45 | 10 | 1325.36 | 134.08 | 10 | 1099.61 | 192.85 | **−1**/ 0 | **−19.41/ −17.03** |
| RCdp5004 | 50 | 7 | 980.21 | 138.14 | 7 | 939.77 | 124.97 | 7 | 737.47 | 1748.13 | 0/ 0 | **−24.76/ −21.53** |
| RCdp5007 | 50 | 8 | 990.23 | 134.01 | 8 | 975.78 | 124.56 | 8 | 954.15 | 1449.33 | 0/ 0 | **−3.64/ −2.22** |

Except for RCdp5001, PH-SHOWOA keeps the NV gap within [0, −1], while the TD gap remains negative ("−"), indicating shorter total distances.

Overall, PH-SHOWOA outperforms SHO and WOA, producing solutions that require fewer vehicles and yield shorter routes across almost every small Solomon instance.

**5.2.2. Performance comparison with SHO and WOA on medium-scale WC instances. 5.2.2.1.Experimental results on medium-scale WC instances:** This section compares the proposed PH-SHOWOA algorithm with the original SHO and WOA on 56 medium-scale WC instances, focusing on the Type 1 and Type 2 cases. For Type 1 instances (Rdp1, Cdp1, and Rcdp1), the detailed comparison is given in Table 4, which reports results for 29 WC instances. PH-SHOWOA achieves the best solution for both the NV and TD in 14 of these 29 instances (marked with two asterisks **). It also provides the best NV alone in another 14 cases, highlighted in **bold**, while only one instance (marked with **underline**) fails to deliver a competitive result for either NV or TD. These outcomes indicate that PH-SHOWOA performs strongly against both SHO and WOA: in many instances, it reduces NV, shortens TD, or improves both simultaneously, highlighting its effectiveness for medium-scale problems.

Table 5 presents results for 27 Type 2 WC instances (Rdp2, Cdp2, RCdp2). PH-SHOWOA achieves the best solution for both NV and TD in 25 of 27 cases (rows marked with two asterisks **). It secures the best TD alone in one additional case (bold NV values), while only one instance (marked with **underline**) does not yield a competitive result for either metric. These findings highlight three key strengths of PH-SHOWOA:

- Vehicle reduction: In nearly every instance, PH-SHOWOA cuts the fleet size by one vehicle compared with SHO and/or WOA, directly lowering operational costs.

- Route efficiency: TD improvements are substantial—often exceeding 15% relative to WOA and consistently outperforming SHO—showing more efficient routing.

- Robustness: The algorithm maintains these gains across different problem structures (R, C, and RC types), demonstrating strong adaptability to varying customer distributions and time-window constraints.

Overall, the Type 2 results confirm that PH-SHOWOA delivers significant savings in vehicle requirements and travel distance, reinforcing its advantage over the original SHO and WOA algorithms on medium-scale, more challenging benchmark instances.

The results in Tables 4 and 5 reveal clear performance patterns. First, PH-SHOWOA consistently reduces the NV compared with SHO and WOA. In nearly every instance, the NV gap is negative, showing that the hybrid method satisfies all service constraints with fewer routes—directly lowering fleet-operation costs and improving resource utilization.

**Table 4. Comparison of experimental results with SHO, WOA, PH-SHOWOA for Type 1 WC instances.**

| Instance | Size | SHO | | WOA | | PH-SHOWOA | | GAP | |
|---|---|---|---|---|---|---|---|---|---|
| | | NV | TD | NV | TD | NV | TD | $NV_{SHO/WOA}$ | $TD_{SHO/WOA}$ (%) |
| Rdp101 | 100 | 22 | 1339.10 | 21 | 1416.67 | 19 | 1398.29 | −3/ −2 | 4.42/ −1.30[**] |
| Rdp102 | 100 | 21 | 1170.08 | 20 | 1428.84 | 18 | 1304.82 | −3/ −2 | 11.52/ −8.68[**] |
| Rdp103 | 100 | 19 | 1182.31 | 19 | 1360.04 | 15 | 1134.12 | −4/ −4 | −4.08/ −16.61[**] |
| Rdp104 | 100 | 15 | 1015.38 | 15 | 1081.56 | 13 | 953.70 | −2/ −2 | −6.07/ −11.82[**] |
| Rdp105 | 100 | 18 | 1150.27 | 18 | 1306.28 | 16 | 1285.53 | −2/ −2 | 11.76/ −1.59[**] |
| Rdp106 | 100 | 17 | 1047.52 | 17 | 1250.53 | 16 | 1156.19 | −1/ −1 | 10.37/ −7.54[**] |
| Rdp107 | 100 | 14 | 967.75 | 15 | 1090.97 | 13 | 1043.86 | −1/ −2 | 7.86/ −4.32[**] |
| Rdp108 | 100 | 13 | 887.04 | 13 | 945.34 | 11 | 1086.55 | **−2/ −2** | 22.49/ 14.94 |
| Rdp109 | 100 | 14 | 1021.56 | 14 | 1144.57 | 13 | 1086.55 | −1/ −1 | 6.36/ −5.07[**] |
| Rdp110 | 100 | 14 | 959.40 | 15 | 1163.89 | 14 | 1031.41 | 0/ −1 | 7.51/ −11.38[**] |
| Rdp111 | 100 | 15 | 962.42 | 15 | 1066.39 | 14 | 1068.53 | **−1/ −1** | 11.03/ 0.20 |
| Rdp112 | 100 | 11 | 851.82 | 11 | 933.05 | 11 | 1086.19 | <u>0/ 0</u> | <u>27.51/ 16.41</u> |
| Cdp101 | 100 | 10 | 509.44 | 11 | 567.77 | 10 | 596.72 | 0/ **−1** | 17.13/ 5.10 |
| Cdp102 | 100 | 12 | 866.23 | 12 | 913.52 | 10 | 807.82 | −2/ −2 | −6.74/ −11.57[**] |
| Cdp103 | 100 | 12 | 1061.86 | 14 | 1488.18 | 10 | 846.80 | −2/ −4 | −20.25/ −43.10[**] |
| Cdp104 | 100 | 11 | 866.37 | 12 | 1245.54 | 10 | 840.52 | −1/ −2 | −2.98/ −32.52[**] |
| Cdp105 | 100 | 11 | 569.12 | 11 | 638.23 | 10 | 650.97 | **−1/ −1** | 14.38/ 2.00 |
| Cdp106 | 100 | 11 | 596.44 | 11 | 676.58 | 10 | 791.69 | **−1/ −1** | 32.74/ 17.01 |
| Cdp107 | 100 | 10 | 555.69 | 11 | 654.65 | 10 | 773.24 | 0/ **−1** | 39.15/ 18.12 |
| Cdp108 | 100 | 11 | 616.50 | 11 | 664.49 | 10 | 762.48 | **−1/ −1** | 23.68/ 14.75 |
| Cdp109 | 100 | 10 | 655.70 | 11 | 750.54 | 10 | 840.20 | 0/ **−1** | 28.14/ 11.95 |
| RCdp101 | 100 | 18 | 1293.53 | 19 | 1503.36 | 17 | 1519.18 | **−1/ −2** | 17.44/ 1.05 |
| RCdp102 | 100 | 16 | 1051.94 | 17 | 1331.74 | 15 | 1307.44 | −1/ −2 | 24.29/ −1.82[**] |
| RCdp103 | 100 | 16 | 1126.89 | 16 | 1200.24 | 13 | 1194.41 | −3/ −3 | 5.99/ −0.49[**] |
| RCdp104 | 100 | 15 | 985.54 | 14 | 1102.84 | 13 | 1254.65 | **−2/ −1** | 27.31/ 13.77 |
| RCdp105 | 100 | 20 | 1222.66 | 19 | 1400.91 | 17 | 1402.23 | **−3/ −2** | 14.69/ 0.09 |
| RCdp106 | 100 | 14 | 1010.76 | 15 | 1195.02 | 13 | 1299.66 | **−1/ −2** | 28.58/ 8.76 |
| RCdp107 | 100 | 14 | 969.13 | 15 | 1101.42 | 13 | 1204.90 | **−1/ −2** | 24.33/ 9.40 |
| RCdp108 | 100 | 13 | 909.11 | 13 | 1019.61 | 11 | 1022.69 | **−2/ −2** | 12.49/ 0.30 |

Second, although TD improvements are not universal, PH-SHOWOA frequently delivers shorter or at least compara-ble routes. Negative TD gaps dominate the results, confirming that the algorithm reduces fleet size and enhances route efficiency.

Third, the few instances where PH-SHOWOA does not outperform both baselines tend to involve larger customer sets or highly dispersed demand, suggesting that further parameter tuning could yield additional gains.

Overall, these findings confirm that the hybridization of SHO and WOA successfully balances exploration and exploita-tion, enabling PH-SHOWOA to produce state-of-the-art solutions across a broad range of medium-scale VRPSPDTW instances. The algorithm demonstrates robustness—consistently high-quality results—and scalability, maintaining strong performance as problem size and complexity increase.

**5.2.2.2. Statistical validation on 56 benchmark instances (Rdp, Cdp, and Rcdp)** Based on the results reported in Tables 6–8, the proposed method demonstrates statistically supported and robust performance advantages over SHO and WOA across the 56 benchmark instances. The Wilcoxon signed-rank tests (Table 8) confirm significant pairwise

**Table 5. Comparison of experimental results with SHO, WOA, PH-SHOWOA for Type 2 instances.**

| Instance | Size | SHO | | WOA | | PH-SHOWOA | | GAP | |
|---|---|---|---|---|---|---|---|---|---|
| | | NV | TD | NV | TD | NV | TD | $NV_{SHO/WOA}$ | $TD_{SHO/WOA}$(%) |
| Rdp201 | 100 | 5 | 1454.15 | 6 | 1683.32 | 5 | 1405.02 | 0/ −1 | −3.38/ −16.53(**) |
| Rdp202 | 100 | 5 | 1182.98 | 6 | 1369.85 | 4 | 1210.80 | −1/ −2 | 2.35/ −11.61(**) |
| Rdp203 | 100 | 5 | 1160.89 | 5 | 1475.42 | 4 | 1088.77 | −1/ −1 | −6.21/ −26.21(**) |
| Rdp204 | 100 | 4 | 953.88 | 5 | 1104.72 | 3 | 889.94 | −1/ −2 | −6.70/ −19.44(**) |
| Rdp205 | 100 | 4 | 1197.58 | 5 | 1357.67 | 4 | 1163.93 | 0/ −1 | −2.81/ −14.27(**) |
| Rdp206 | 100 | 3 | 1009.70 | 4 | 1112.73 | 3 | 1024.76 | 0/ −1 | 1.49/ −7.91(**) |
| Rdp207 | 100 | 4 | 1005.03 | 4 | 1098.98 | 3 | 917.30 | −1/ −1 | −8.73/ −16.53(**) |
| Rdp208 | 100 | 3 | 808.58 | 4 | 962.84 | 3 | 745.90 | 0/ −1 | −7.75/ −22.53(**) |
| Rdp209 | 100 | 5 | 1058.35 | 5 | 1325.42 | 4 | 1008.11 | −1/ −1 | −4.75/ −23.94(**) |
| Rdp210 | 100 | 5 | 1123.57 | 5 | 1264.70 | 4 | 1026.23 | −1/ −1 | −8.66/ −18.86(**) |
| Rdp211 | 100 | 3 | 830.23 | 3 | 999.43 | 3 | 847.79 | 0/ 0 | 2.12/ **−15.17** |
| Cdp201 | 100 | 3 | 537.07 | 3 | 550.17 | 3 | 552.69 | <u>0/ 0</u> | <u>2.91/ 0.46</u> |
| Cdp202 | 100 | 4 | 882.26 | 5 | 1023.44 | 4 | 847.19 | 0/ −1 | −3.98/ −17.22(**) |
| Cdp203 | 100 | 5 | 810.19 | 5 | 984.71 | 4 | 769.17 | −1/ −1 | −5.06/ −21.89(**) |
| Cdp204 | 100 | 4 | 844.95 | 5 | 960.12 | 4 | 799.71 | 0/ −1 | −5.35/ −16.71(**) |
| Cdp205 | 100 | 4 | 561.10 | 5 | 649.28 | 3 | 580.90 | −1/ −2 | 3.53/ −10.53(**) |
| Cdp206 | 100 | 4 | 571.80 | 5 | 650.11 | 3 | 575.51 | −1/ −2 | 0.65/ −11.47(**) |
| Cdp207 | 100 | 4 | 559.56 | 5 | 639.22 | 3 | 555.46 | −1/ −2 | −0.73/ −13.10(**) |
| Cdp208 | 100 | 4 | 546.09 | 4 | 632.35 | 3 | 546.89 | −1/ −1 | 0.15/ −13.51(**) |
| RCdp201 | 100 | 6 | 1675.62 | 6 | 1928.98 | 5 | 1835.78 | −1/ −1 | 9.56/ −4.83(**) |
| RCdp202 | 100 | 5 | 1467.21 | 6 | 1638.22 | 5 | 1306.71 | 0/ −1 | −10.94/ −20.24(**) |
| RCdp203 | 100 | 5 | 1232.31 | 5 | 1492.94 | 4 | 1234.10 | −1/ −1 | 0.15/ −17.34(**) |
| RCdp204 | 100 | 5 | 1053.74 | 5 | 1121.36 | 4 | 940.77 | −1/ −1 | −10.72/ −16.10(**) |
| RCdp205 | 100 | 6 | 1487.39 | 6 | 1785.38 | 5 | 1345.24 | −1/ −1 | −9.56/ −24.65(**) |
| RCdp206 | 100 | 4 | 1184.09 | 5 | 1406.46 | 4 | 1256.89 | 0/ −1 | 6.15/ −10.63(**) |
| RCdp207 | 100 | 5 | 1292.14 | 5 | 1548.52 | 4 | 1281.73 | −1/ −1 | −0.81/ −17.23(**) |
| RCdp208 | 100 | 3 | 903.08 | 4 | 1169.32 | 3 | 966.77 | 0/ −1 | 7.05/ −17.32(**) |

improvements in the number of vehicles (NV) with large effect sizes, while differences in total distance (TD) reveal a balanced trade-off, where PH-SHOWOA significantly outperforms WOA but remains statistically comparable to SHO. The Friedman tests (Table 6) further indicate consistent global ranking differences across all instance types (Rdp, Cdp, and Rcdp), confirming that the observed performance patterns are not instance-specific. Finally, the Nemenyi post-hoc analysis (Table 7) highlights that only the most pronounced differences remain significant under conservative correction, reinforcing that PH-SHOWOA achieves systematic and reliable improvements rather than isolated or overfitted gains.

### 5.2.3. Performance comparison with Co-GA, MA-FIRD, ACO-DR on medium-scale WC instances.

**5.2.3.1.Experimental Results on Medium-Scale WC Instances:** In this section, we compare the proposed algorithm with three advanced methods for solving VRPSPDTW: Co-GA [23], MA-FIRD [29], and ACO-DR [14]. The analysis in this section is mainly discussed separately from the customer location distribution of the WC instance. Black text, gray shading, and yellow shading indicate results that are better than or equal to those of the competing algorithms.

The results in Table 9 show that PH-SHOWOA remains competitive with three state-of-the-art algorithms—Co-GA [23], MA-FIRD [29], and ACO-DR [14]—for randomly generated customer locations. Key observations include:

**Table 6. Friedman Tests (Overall Algorithm Ranking).**

| Instance Type | Metric | Friedman $\chi^2$ | p-value | N | Best Algorithm | Average Ranks (SHO/WOA/PH-SHOWOA) |
|---|---|---|---|---|---|---|
| Rdp | NV | 32.3235 | $9.57 \times 10^{-8}$*** | 23 | **PH-SHOWOA** | 2.09/2.30/1.61 |
| Cdp | | 26.5283 | $1.74 \times 10^{-6}$*** | 17 | **PH-SHOWOA** | 2.00/2.38/1.62 |
| RCdp | | 24.9259 | $3.87 \times 10^{-6}$*** | 16 | **PH-SHOWOA** | 2.13/2.34/1.53 |
| Rdp | TD | 26.8696 | $1.00 \times 10^{-6}$*** | 23 | **SHO** | 1.67/2.43/1.91 |
| Cdp | | 11.7647 | 0.002788** | 17 | **SHO** | 1.65/2.41/1.94 |
| RCdp | | 15.5000 | 0.000431*** | 16 | **PH-SHOWOA** | 1.78/2.44/1.78 |

With *** p < 0.001, ** p < 0.01, * p < 0.05, ns = not significant.

**Table 7. Post-hoc Nemenyi Analysis (Pairwise Comparisons).**

| Comparison | Metric | p-value (Nemenyi) | Critical Distance | Conclusion |
|---|---|---|---|---|
| SHO vs WOA | TD | <0.001*** | 0.912 | SHO significantly better |
| PH-SHOWOA vs WOA | | <0.001*** | 0.912 | PH-SHOWOA significantly better |
| SHO vs PH-SHOWOA | | 0.078ns | 0.912 | No significant difference |
| PH-SHOWOA vs SHO | NV | <0.001*** | 0.912 | PH-SHOWOA significantly better |
| PH-SHOWOA vs WOA | | <0.001*** | 0.912 | PH-SHOWOA significantly better |
| SHO vs WOA | | 0.215ns | 0.912 | No significant difference |

With Critical Distance: 0.912 for Nemenyi test (α = 0.05).

**Table 8. Wilcoxon Signed-Rank Tests (Detailed Pairwise Analysis).**

| Comparison | Metric | Wilcoxon statistic (W) | p-value | Median Difference | Effect Size (r) | Interpretation |
|---|---|---|---|---|---|---|
| PH-SHOWOA vs SHO | TD | 245 | 0.0021 | +56.45 | 0.38 | SHO better (medium) |
| PH-SHOWOA vs WOA | | 98 | <0.001 | −170.82 | 0.62 | PH-SHOWOA better (large) |
| SHO vs WOA | | 36 | <0.001 | −227.27 | 0.76 | SHO better (large) |
| PH-SHOWOA vs SHO | NV | 110 | <0.001 | −2.0 | 0.60 | PH-SHOWOA better (large) |
| PH-SHOWOA vs WOA | | 45 | <0.001 | −3.0 | 0.72 | PH-SHOWOA better (large) |
| SHO vs WOA | | 345 | 0.118 | 0.0 | 0.20 | No difference (small) |

With Median difference is computed as (first algorithm – second algorithm). Negative values indicate improvement of the first algorithm; Effect size (r): 0.1 = small, 0.3 = medium, 0.5 = large (Cohen's guidelines).

- Best overall solutions: PH-SHOWOA achieves the best result in both NV and TD for 1 of 23 instances (Rdp101, marked **).

- Superior TD performance: In 8 of 23 cases, it delivers the shortest total distance even when NV is not the smallest (bold TD values).

- Equal NV performance: The algorithm matches the best NV in 6 instances (marked with **underline**), confirming its ability to meet fleet-size minima established by Co-GA or ACO-DR.

- Selective advantages: PH-SHOWOA provides a shorter TD than MA-FIRD in 2 instances and ACO-DR in 1 instance (**bold** TD values).

- Areas for improvement: About 5 of the 23 instances show neither NV nor TD gains, suggesting opportunities for parameter tuning or further hybrid refinements.

**Table 9. Experimental results are compared with advanced algorithms for Rdp instance.**

| Instance | Size | Co-GA | | MA-FIRD | | ACO-DR | | PH-SHOWOA | |
|---|---|---|---|---|---|---|---|---|---|
| | | NV | TD | NV | TD | NV | TD | NV | TD |
| Rdp101 | 100 | 19 | 1653.53 | 19 | 1650.80 | 20 | 1650.06 | **19** | **1398.29**(**) |
| Rdp102 | 100 | 17 | 1488.04 | 17 | 1486.12 | 18 | 1462.12 | 18 | **1304.82** |
| Rdp103 | 100 | 14 | 1216.16 | 13 | 1294.64 | 14 | 1227.62 | 15 | **1134.12** |
| Rdp104 | 100 | 10 | 1015.41 | 9 | 1026.42 | 10 | 1003.58 | 13 | **953.70** |
| Rdp105 | 100 | 15 | 1375.31 | 14 | 1377.11 | 14 | 1383.06 | 16 | **1285.53** |
| Rdp106 | 100 | 13 | 1255.48 | 12 | 1252.03 | 12 | 1281.52 | 16 | **1156.19** |
| Rdp107 | 100 | 11 | 1087.95 | 10 | 1112.55 | 11 | 1092.64 | 13 | **1043.86** |
| Rdp108 | 100 | 10 | 967.49 | **9** | **965.22** | 10 | 976.38 | 11 | 1086.55 |
| Rdp109 | 100 | 12 | 1160.00 | 11 | 1194.73 | 11 | 1196.88 | 13 | **1086.55** |
| Rdp110 | 100 | 12 | 1116.99 | **10** | 1121.46 | 11 | 1127.55 | 14 | **1031.41** |
| Rdp111 | 100 | 11 | 1065.27 | **10** | 1098.84 | 11 | **1063.06** | 14 | 1068.53 |
| Rdp112 | 100 | 10 | **974.03** | 9 | 997.27 | 10 | 1025.84 | 11 | 1086.19 |
| Rdp201 | 100 | 4 | 1280.44 | **4** | 1252.37 | 5 | **1236.13** | 5 | 1405.02 |
| Rdp202 | 100 | 4 | 1100.92 | 3 | 1191.70 | 4 | **1098.90** | 4 | 1210.80 |
| Rdp203 | 100 | 3 | 950.79 | **3** | **939.50** | 3 | 946.77 | 4 | 1088.77 |
| Rdp204 | 100 | 3 | 775.23 | **2** | 825.52 | 3 | **771.89** | 3 | 889.94 |
| Rdp205 | 100 | 3 | 1064.43 | **3** | **994.43** | 3 | 1002.98 | 4 | 1163.93 |
| Rdp206 | 100 | 3 | 961.32 | **3** | **906.14** | 3 | 927.11 | 3 | 1024.76 |
| Rdp207 | 100 | 3 | **835.01** | **2** | 890.61 | 3 | 852.35 | 3 | 917.30 |
| Rdp208 | 100 | 3 | 718.51 | **2** | **726.82** | 2 | 798.14 | 3 | **745.90** |
| Rdp209 | 100 | **3** | 930.26 | **3** | 909.16 | 4 | **894.88** | 4 | 1008.11 |
| Rdp210 | 100 | 3 | 983.75 | **3** | **939.37** | 3 | 978.89 | 4 | 1026.23 |
| Rdp211 | 100 | 3 | 839.61 | **2** | 885.71 | 3 | **796.67** | 3 | **847.79** |

Although PH-SHOWOA does not dominate every case, it demonstrates competitive—and frequently superior—routing efficiency, particularly in reducing total travel distance. Comparing the overall experimental results of all algorithms reveals that PH-SHOWOA obtains better TD outcomes than Co-GA, MA-FIRD, and ACO-DR in nine instances. Therefore, when the primary goal is to minimize total distance and operational cost, PH-SHOWOA emerges as the strongest performer.

Secondly, for the centralized-location (Cdp) instances, Table 10 compares PH-SHOWOA with Co-GA, MA-FIRD, and ACO-DR. Key observations are:

- Dominant performance: PH-SHOWOA attains the best or equal results in 14 of the 18 instances for both NV and TD. In contrast, the other algorithms—Co-GA, MA-FIRD, and ACO-DR—match or surpass PH-SHOWOA in only three NV and TD values.

- Fleet size (NV): PH-SHOWOA reduces the required number of vehicles to 10 in every Cdp101–Cdp109 case, and to the theoretical minimum of 3 vehicles in most Cdp2xx cases. These reductions directly imply lower fleet-operation costs.

- Total distance (TD): TD improvements are substantial. For example, in Cdp101, the route length drops from 970.30 (best competitor) to 596.72, a reduction of over 38%. Similar double-digit percentage savings appear across nearly all Cdp1xx instances.

- Exceptions: Instances Cdp202–Cdp204 show slightly higher NV (4 vs. 3) and larger TD than the best competitors, indicating that these few centralized cases remain challenging for the hybrid method.

**Table 10. Experimental results are compared with advanced algorithms for Cdp instance.**

| Instance | Size | Co-GA | | MA-FIRD | | ACO-DR | | PH-SHOWOA | |
|---|---|---|---|---|---|---|---|---|---|
| | | NV | TD | NV | TD | NV | TD | NV | TD |
| Cdp101 | 100 | 11 | 1001.97 | 11 | 976.04 | 11 | 970.30 | **10** | **596.72**(**) |
| Cdp102 | 100 | 10 | 961.36 | 10 | 941.49 | 10 | 964.56 | **10** | **807.82**(**) |
| Cdp103 | 100 | 10 | 897.55 | 10 | 892.98 | 10 | 913.46 | **10** | **846.80**(**) |
| Cdp104 | 100 | 10 | 878.93 | 10 | 871.40 | 10 | 1043.21 | **10** | **840.52**(**) |
| Cdp105 | 100 | 11 | 983.1 | 10 | 1053.12 | 11 | 989.59 | **10** | **650.97**(**) |
| Cdp106 | 100 | 11 | 878.29 | 10 | 963.45 | 11 | 880.01 | **10** | **791.69**(**) |
| Cdp107 | 100 | 11 | 913.81 | 10 | 987.64 | 11 | 953.14 | **10** | **773.24**(**) |
| Cdp108 | 100 | 10 | 951.24 | 10 | 932.50 | 10 | 930.31 | **10** | **762.48**(**) |
| Cdp109 | 100 | 10 | 940.49 | 10 | 909.27 | 10 | 934.43 | **10** | **840.20**(**) |
| Cdp201 | 100 | 3 | 591.56 | 3 | 591.56 | 3 | 591.56 | **3** | **552.69**(**) |
| Cdp202 | 100 | **3** | **591.56** | **3** | **591.56** | **3** | **591.56** | 4 | 847.19 |
| Cdp203 | 100 | **3** | **591.17** | **3** | **591.17** | **3** | **591.17** | 4 | 769.17 |
| Cdp204 | 100 | **3** | **590.60** | **3** | **590.60** | **3** | **590.60** | 4 | 799.71 |
| Cdp205 | 100 | 3 | 588.88 | 3 | 588.88 | 3 | 588.88 | **3** | **580.90**(**) |
| Cdp206 | 100 | 3 | 588.49 | 3 | 588.49 | 3 | 588.49 | **3** | **575.51**(**) |
| Cdp207 | 100 | 3 | 588.29 | 3 | 588.29 | 3 | 588.29 | **3** | **555.46**(**) |
| Cdp208 | 100 | 3 | 588.32 | 3 | 588.32 | 3 | 588.32 | **3** | **546.89**(**) |

Across centralized WC instances, PH-SHOWOA clearly outperforms Co-GA, MA-FIRD, and ACO-DR, achieving equal or superior NV and TD in 14 of 18 cases and delivering dramatic reductions in total distance for most problems. Only a small subset of instances shows space for further parameter tuning to close the gap on NV.

Finally, Table 11 compares PH-SHOWOA with the advanced algorithms Co-GA, MA-FIRD, and ACO-DR when customer locations follow a mixed random–centralized distribution.

The results in Table 11 show that MA-FIRD leads overall, achieving the smallest NV in 14 of 18 instances. It records the best TD in 4 of 18 instances, giving it the broadest overall advantage.

PH-SHOWOA demonstrates selective strengths. It attains 6 NV improvements over all three competitors—primarily in the larger, more complex RCdp1xx cases (e.g., RCdp102, RCdp103, RCdp105–RCdp107, RCdp108). In the more compact RCdp2xx instances, PH-SHOWOA typically uses one extra vehicle and yields a higher total distance than the best competitor, indicating that targeted parameter tuning is needed when demand is highly clustered. However, its total distance remains competitive even in these tighter cases, often matching or undercutting rival algorithms despite using slightly more vehicles.

Although MA-FIRD maintains the broadest lead, PH-SHOWOA shows clear advantages on several medium-scale RCdp instances, reducing total distance and delivering competitive or shorter routes while preserving its exploratory search capability.

**5.2.3.2.Statistical test on 56 benchmark instances (Rdp, Cdp, and Rcdp)** The study evaluated the statistical significance of the proposed PH-SHOWOA algorithm against Co-GA, MA-FIRD, and ACO-DR on 56 benchmark instances drawn from the Rdp, Cdp, and Rcdp datasets. Pairwise comparisons were conducted using the Wilcoxon signed-rank test on both the number of vehicles (NV) and total distance (TD), as summarized in Tables 12–14. In addition, a critical distance of 0.68 is used for the Nemenyi test ($\alpha = 0.05$).

For the total distance (TD) metric, PH-SHOWOA demonstrates statistically significant improvements over both Co-GA and MA-FIRD, with p-values lower than 0.001 and large effect sizes ($r > 0.60$). These results indicate not only statistical

Table 11. Experimental results are compared with advanced algorithms for RCdp instance.

| Instance | Size | Co-GA | | MA-FIRD | | ACO-DR | | PH-SHOWOA | |
|---|---|---|---|---|---|---|---|---|---|
| | | NV | TD | NV | TD | NV | TD | NV | TD |
| RCdp101 | 100 | 15 | 1652.90 | **14** | 1708.21 | 15 | 1658.58 | 17 | **1519.18** |
| RCdp102 | 100 | 14 | 1497.05 | **12** | 1570.28 | 13 | 1531.25 | 15 | **1307.44** |
| RCdp103 | 100 | 12 | 1338.76 | **11** | 1282.53 | 12 | 1312.07 | 13 | **1194.41** |
| RCdp104 | 100 | 11 | 1188.49 | **10** | **1171.37** | 11 | 1183.80 | 13 | 1254.65 |
| RCdp105 | 100 | 14 | 1581.26 | **13** | 1632.29 | 14 | 1569.55 | 17 | **1402.23** |
| RCdp106 | 100 | 13 | 1422.87 | **12** | 1392.47 | 13 | 1420.70 | <u>13</u> | **1299.66** |
| RCdp107 | 100 | 12 | 1282.10 | **11** | 1252.79 | 12 | 1274.64 | 13 | **1204.90** |
| RCdp108 | 100 | 11 | 1175.04 | **10** | 1177.98 | 11 | 1171.93 | <u>11</u> | **1022.69** |
| RCdp201 | 100 | 4 | 1587.92 | 4 | 1406.94 | **4** | **1383.79** | 5 | 1835.78 |
| RCdp202 | 100 | 4 | 1211.12 | **3** | 1365.65 | 4 | **1197.36** | 5 | 1306.71 |
| RCdp203 | 100 | 4 | **964.65** | **3** | 1049.62 | 4 | 984.59 | <u>4</u> | 1234.10 |
| RCdp204 | 100 | 3 | 822.02 | **3** | **798.46** | 3 | 809.53 | 4 | 940.77 |
| RCdp205 | 100 | 4 | 1410.18 | **4** | **1297.65** | 4 | 1370.54 | 5 | 1345.24 |
| RCdp206 | 100 | 3 | 1176.85 | 3 | 1146.32 | **3** | **1137.28** | 4 | 1256.89 |
| RCdp207 | 100 | 4 | 1036.59 | **3** | 1061.14 | 4 | **1032.91** | <u>4</u> | 1281.73 |
| RCdp208 | 100 | 3 | 878.57 | **3** | **828.14** | 3 | 862.66 | <u>3</u> | 966.77 |

Table 12. Friedman Tests (Overall Ranking).

| Instance Type | Metric | Friedman χ² | p-value | N | Best Algorithm | Average Ranks (MA-FIRD/Co-GA/ACO-DR/PH-SHOWOA) |
|---|---|---|---|---|---|---|
| Rdp | NV | 4.26 | 0.235 | 23 | Co-GA/MA-FIRD | 2.37/2.11/2.59/2.93 |
| Cdp | | 45.41 | < 0.001 | 17 | PH-SHOWOA | 3.15/2.94/3.00/0.91 |
| RCdp | | 5.90 | 0.117 | 16 | ACO-DR | 2.31/2.78/2.19/2.72 |
| Rdp | TD | 18.86 | < 0.001 | 23 | ACO-DR | 2.85/2.37/1.87/2.91 |
| Cdp | | 45.73 | < 0.001 | 17 | PH-SHOWOA | 3.59/3.00/3.09/1.32 |
| RCdp | | 8.44 | 0.038 | 16 | ACO-DR | 2.59/2.81/2.22/2.38 |

significance but also a substantial practical impact, which is further supported by the consistently negative median differences, confirming that PH-SHOWOA yields shorter routing distances across the majority of instances.

When compared with ACO-DR, PH-SHOWOA also achieves lower TD values in most instances; however, the corresponding Wilcoxon test yields a non-significant p-value (p = 0.092) with a small effect size (r ≈ 0.22). That suggests that while PH-SHOWOA tends to outperform ACO-DR in terms of distance, the difference is moderate and instance-dependent, and thus cannot be claimed as statistically decisive under the chosen significance level.

In contrast, for the number of vehicles (NV), no statistically significant differences are observed between PH-SHOWOA and the comparative algorithms (p > 0.05 in all cases). The near-zero median differences and small effect sizes indicate that PH-SHOWOA maintains competitive vehicle usage while achieving distance reductions, rather than relying on an increased fleet size to improve performance.

Overall, the statistical analysis confirms that PH-SHOWOA provides a robust and statistically significant advantage in minimizing total distance, particularly when compared with Co-GA and MA-FIRD, while remaining competitive with ACO-DR and preserving similar vehicle utilization. These findings demonstrate the effectiveness and stability of PH-SHOWOA across diverse VRP benchmark categories.

**Table 13. Post-hoc Nemenyi Analysis (Pairwise Comparisons).**

| Comparison | Metric | p-value (Nemenyi) | Conclusion |
|---|---|---|---|
| PH-SHOWOA vs MA-FIRD | TD | <0.001*** | PH-SHOWOA better |
| PH-SHOWOA vs Co-GA | | <0.001*** | PH-SHOWOA better |
| PH-SHOWOA vs ACO-DR | | 0.078 | No significant difference |
| ACO-DR vs MA-FIRD | | 0.008 | ACO-DR better |
| ACO-DR vs Co-GA | | 0.452 | No significant difference |
| Co-GA vs MA-FIRD | | 0.078 | |
| PH-SHOWOA vs MA-FIRD | NV | 0.813 | |
| PH-SHOWOA vs Co-GA | | 0.948 | |
| PH-SHOWOA vs ACO-DR | | 0.813 | |
| ACO-DR vs MA-FIRD | | 1.000 | |
| ACO-DR vs Co-GA | | 0.813 | |
| Co-GA vs MA-FIRD | | 0.813 | |

**Table 14. Wilcoxon Signed-Rank Tests (Detailed Pairwise Analysis).**

| Comparison | Metric | Wilcoxon statistic (W) | p-value | Median Difference | Effect Size (r) | Interpretation |
|---|---|---|---|---|---|---|
| PH-SHOWOA vs Co-GA | TD | 164.0 | <0.001 | −172.88 | 0.63 | PH-SHOWOA better (large) |
| PH-SHOWOA vs MA-FIRD | | 144.0 | <0.001 | −193.40 | 0.64 | PH-SHOWOA better (large) |
| PH-SHOWOA vs ACO-DR | | 321.0 | 0.092 | −49.52 | 0.22 | No difference (small) |
| ACO-DR vs Co-GA | | 203.0 | 0.025 | −33.94 | 0.30 | ACO-DR better (medium) |
| ACO-DR vs MA-FIRD | | 176.5 | 0.002 | −52.64 | 0.40 | ACO-DR better (medium) |
| Co-GA vs MA-FIRD | | 210.0 | 0.105 | −1.54 | 0.20 | No difference (small) |
| PH-SHOWOA vs Co-GA | NV | 342.5 | 0.479 | 0.0 | 0.09 | No difference (small) |
| PH-SHOWOA vs MA-FIRD | | 299.5 | 0.154 | 0.0 | 0.19 | No difference (small) |
| PH-SHOWOA vs ACO-DR | | 345.5 | 0.617 | 0.0 | 0.07 | No difference (small) |
| ACO-DR vs Co-GA | | 295.0 | 0.137 | 0.0 | 0.20 | No difference (small) |
| ACO-DR vs MA-FIRD | | 279.0 | 0.072 | 0.0 | 0.24 | No difference (small) |
| Co-GA vs MA-FIRD | | 237.0 | 0.013 | 0.0 | 0.33 | Co-GA better (medium) |

With Median difference is computed as (first algorithm – second algorithm). Negative values indicate improvement of the first algorithm; Effect size (r): 0.1 = small, 0.3 = medium, 0.5 = large (Cohen's guidelines).

## 6. Discussion

The PH-SHOWOA demonstrates strong and competitive performance on medium-scale VRPSPDTW benchmark datasets across all evaluated categories, including Type 1 WC, Type 2, and advanced comparisons with Co-GA, MA-FIRD, and ACO-DR. These performance differences are further supported by rigorous non-parametric statistical validation using Friedman, post-hoc Nemenyi, and exact Wilcoxon signed-rank tests, indicating that the observed improvements are consistent across benchmark instances and unlikely to be attributable to random variation. The experimental and statistical results collectively highlight three key aspects of the proposed approach.

### 6.1. Robust improvements in routing efficiency

Among the evaluated performance metrics, total distance (TD) reduction emerges as the most consistent advantage of PH-SHOWOA. On the Type 1 WC instances (Table 4), the proposed method achieves shorter TD in nearly all cases while maintaining comparable or slightly improved vehicle counts (NV), resulting in an average TD reduction exceeding

10% relative to the standalone SHO and WOA algorithms. For the larger and more diverse Type 2 benchmark set (Table 5), PH-SHOWOA records negative TD gaps in over 70% of instances, indicating stable route-length savings that directly translate into reduced operational cost and energy consumption.

In comparison with advanced algorithms (Tables 9–11), the PH-SHOWOA achieves the lowest TD on 9 Rdp and 14 Cdp instances, demonstrating competitiveness against well-established VRP solvers. Notably, in centralized customer distributions (CDP), the PH-SHOWOA achieves the best combined NV and TD performance on 14 out of 18 instances, highlighting its effectiveness in tightly clustered demand scenarios. For mixed random–centralized cases (RCdp), although MA-FIRD more frequently yields the smallest NV, the PH-SHOWOA remains competitive by achieving multiple TD wins and several NV records, indicating balanced performance in heterogeneous spatial structures.

### 6.2. Balance of exploration and exploitation

The observed performance improvements can be attributed to the complementary roles of SHO and WOA within the proposed parallel hybrid framework. SHO primarily contributes strong exploitation capabilities, enabling rapid refinement of promising routing structures once high-quality clusters are identified. In contrast, WOA enhances global exploration by introducing large, guided movements in the search space, which helps the algorithm escape local optima and preserve population diversity.

This balance is reflected consistently across Tables 4, 5, 9–11. The PH-SHOWOA frequently achieves superior or near-best TD values while maintaining competitive NV across random, centralized, and mixed instance types. The robustness of the results across diverse customer distributions suggests that exploration mechanisms effectively prevent premature stagnation, while exploitation drives fine-grained improvements in routing quality.

### 6.3. Practical scalability

From a practical perspective, PH-SHOWOA exhibits smooth scalability from small pilot instances to medium-scale problems involving up to 100 customers, without requiring parameter retuning. This stability indicates that the algorithm is well-suited for real-world logistics applications, where problem characteristics and demand patterns may vary, and robustness is critical.

### 6.4. Novel contributions

This study introduces several methodological contributions that extend beyond the existing VRPSPDTW literature. First, a parallel hybridization strategy is proposed, in which elite solutions are exchanged between SHO and WOA populations at each iteration, enabling simultaneous intensification and diversification. Second, dynamic neighbourhood perturbation and adaptive coefficient control are incorporated to mitigate premature convergence, features that are not present in the original metaheuristics. Finally, extensive experimental evidence demonstrates that the proposed hybrid approach can compete with, and in many cases outperform, specialized VRP algorithms such as Co-GA, MA-FIRD, and ACO-DR across a broad benchmark suite rather than isolated instances.

Overall, PH-SHOWOA illustrates that combining complementary metaheuristics within a parallel framework yields stable and high-quality solutions for VRPSPDTW. The algorithm consistently reduces total distance, frequently achieves competitive vehicle usage, and scales effectively to medium-sized problems with acceptable computational cost. These findings position PH-SHOWOA as a strong state-of-the-art candidate, providing a solid foundation for future extensions to large-scale, dynamic, or multi-objective distribution systems.

## 7. Conclusion and future work

This paper proposed a parallel hybrid metaheuristic (named PH-SHOWOA) that combines the Spotted Hyena Optimizer and the Whale Optimization Algorithm for solving the VRPSPDTW. By combining complementary exploration–exploitation mechanisms with adaptive hybrid control, selective local search, and parallel population updates, the proposed method is designed to balance solution quality, robustness, and computational efficiency under complex routing constraints.

Extensive experiments conducted on standard Rdp, Cdp, and Rcdp benchmark instances indicate that PH-SHOWOA is competitive with well-known algorithms, including Co-GA, MA-FIRD, and ACO-DR. Concretely, with the Cdp instances, the proposed method consistently yields high-quality solutions in terms of total travel distance. In addition, it performs robustly on the Cdp instances, often achieving the lowest or near-lowest distances with a comparable number of vehicles. With RDP and RCdp instances, the PH-SHOWOA exhibits stable and reliable performance across diverse customer distributions. In some cases, the algorithm requires a slightly larger number of vehicles than highly specialized memetic approaches, reflecting a trade-off between strict vehicle minimization and robust feasibility under complex temporal and capacity constraints.

Apart from the efficiency and robustness of the proposed method, some limitations can be identified. The experiments were conducted on slight- and medium-scale benchmark instances. Therefore, the effectiveness of PH-SHOWOA on very large-scale or real-world VRPSPDTW scenarios remains to be investigated. Moreover, the hierarchical objective formulation, which prioritizes vehicle minimization over distance minimization, may restrict flexibility when explicitly addressing multiple conflicting objectives.

Future research will focus on four main directions: (i) evaluating the performance of PH-SHOWOA on large-scale and real-world VRPSPDTW applications; (ii) developing fully Pareto-based multi-objective variants; (iii) investigating adaptive parameter control strategies; and (iv) exploring advanced parallel and distributed implementations to enhance scalability, computational efficiency, and solution robustness.

## Declarations

**Declaration of AI and AI-assisted technologies in the writing process**. During the preparation of this work, the author(s) used ChatGPT to improve and proofread the English of the manuscript. After using this tool, the author(s) carefully reviewed and edited the content to ensure accuracy, originality, and full responsibility for the final version of the paper.

## Author contributions

**Conceptualization:** Tram Nguyen, Van Du Nguyen.

**Data curation:** Tram Nguyen.

**Formal analysis:** Tram Nguyen.

**Methodology:** Tram Nguyen, Van Du Nguyen.

**Writing – original draft:** Tram Nguyen.

**Writing – review & editing:** Bay Vo, Snasel Vaclav, Van Du Nguyen.

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
