## [Decision Letter · Decision Letter 0]

17 Dec 2025

Dear Dr. Nguyen,

We look forward to receiving your revised manuscript.

Kind regards,

Nazanin Tajik

Academic Editor

PLOS One

Journal Requirements:

[The authors are grateful to the anonymous referees for carefully reading the manuscript and providing constructive comments, which helped substantially improve the paper. This work was supported by the project “Research and propose methods to solve transport optimization problems,” which received funding from the Information Technology Faculty (FIT), Nong Lam University - Thu Duc City - Ho Chi Minh City - Vietnam, with code project CS-CB23-CNTT-02.]

[The author(s) received no specific funding for this work.]

4. Please note that your Data Availability Statement is currently missing the DOI/accession number of each dataset OR a direct link to access each database. If your manuscript is accepted for publication, you will be asked to provide these details on a very short timeline. We therefore suggest that you provide this information now, though we will not hold up the peer review process if you are unable.

Reviewers' comments:

Reviewer's Responses to Questions

**Comments to the Author**

1. Is the manuscript technically sound, and do the data support the conclusions?

Reviewer #1: Yes

Reviewer #2: Yes

2. Has the statistical analysis been performed appropriately and rigorously?

Reviewer #1: No

Reviewer #2: N/A

3. Have the authors made all data underlying the findings in their manuscript fully available?

Reviewer #1: Yes

Reviewer #2: Yes

4. Is the manuscript presented in an intelligible fashion and written in standard English?

Reviewer #1: Yes

Reviewer #2: Yes

Reviewer #1: The manuscript presents an interesting and potentially useful hybrid metaheuristic (PH‑SHOWOA) for the VRPSPDTW and reports extensive experiments on benchmark instances. The overall structure is clear and the topic fits well within the scope of the journal. However, the paper currently falls short of publication standards and requires substantial revision before it can be considered further.

The main concerns relate to (i) clarity and internal consistency of the problem and algorithmic description, (ii) fairness and completeness of the computational comparisons, and (iii) the strength of the performance and novelty claims. The attached files mentioned the comments.

I encourage the authors to carefully clean and streamline the algorithm descriptions, provide a fully reproducible parameter specification, and, where possible, run competing methods under common experimental conditions or temper the strength of the comparative statements. Adding formal statistical analyses would also greatly strengthen the empirical section. With these issues addressed and the claims aligned more closely with the evidence, the work could make a valuable contribution to the literature on VRPSPDTW solution methods.

Reviewer #2: The paper makes a solid contribution to the VRPSPDTW literature by being the first to apply a hybrid SHO-WOA approach to this problem variant. The methodology is generally sound, combining population-based search with adaptive probability control, simulated annealing acceptance, periodic local search, and parallelization. The experimental evaluation is comprehensive, covering multiple instance categories and comparing against recent competitive algorithms. The paper is reasonably well-written with clear algorithm descriptions that support reproducibility. However, several areas require improvement to strengthen the contribution and meet publication standards. I recommend minor revision to address statistical validation, clarify algorithmic details, and enhance the performance analysis.

The paper has several notable strengths that support its contribution to the field. First, it addresses an important and challenging NP-hard problem with significant practical applications in modern logistics and sustainable transportation. The VRPSPDTW combines multiple real-world constraints, simultaneous pickup and delivery operations with strict time windows, making it highly relevant for contemporary distribution systems.

Second, the methodological novelty of applying SHO and WOA to VRPSPDTW is genuine and meaningful. While these bio-inspired algorithms have been successfully applied to various optimization problems, their combination for this specific routing variant represents new territory. The hybrid framework intelligently leverages SHO's exploitative hunting strategy alongside WOA's exploratory bubble-net behavior, creating a complementary search mechanism.

Third, the experimental evaluation is comprehensive and rigorous in scope. Testing on 65 benchmark instances across different customer distribution patterns (random, clustered, and mixed) provides thorough coverage of problem characteristics. The comparison against three recent state-of-the-art methods (Co-GA from 2012, ACO-DR from 2023, and MA-FIRD from 2024) situates the work appropriately within current research.

Fourth, the results demonstrate clear practical value, particularly for clustered customer scenarios. The algorithm achieves best performance on 14 of 18 Cdp instances for both number of vehicles and total distance, with some solutions showing distance reductions exceeding 30% compared to competitors. Even on more challenging instance types, the method frequently produces competitive or superior total distances, which directly translates to operational cost savings.

Finally, the paper provides detailed algorithm descriptions through eight well-structured pseudocode blocks. This level of detail, including the initialization procedure, hybrid update mechanisms, crossover and mutation operators, and combined local search, supports reproducibility and enables other researchers to build upon this work.

Required Revisions:

- Statistical Validation

The paper currently presents only mean values across 30 independent runs without any statistical significance testing. While the results appear promising, readers cannot determine whether the observed improvements represent genuine algorithmic advantages or merely random variation. This is particularly important given the stochastic nature of metaheuristic algorithms and the relatively small performance differences in some instances.

The authors should add appropriate statistical tests to validate their claims. A Wilcoxon signed-rank test would be suitable for pairwise comparisons between PH-SHOWOA and each baseline algorithm (SHO, WOA, and ideally the competing methods). For comparisons across multiple instance categories, a Friedman test followed by post-hoc analysis would provide more robust validation. Additionally, including standard deviations or confidence intervals in Tables 4-8, or providing box plots for representative instances, would help readers assess solution variability and algorithm stability. A summary table presenting aggregate statistical results across instance categories would strengthen the experimental section considerably.

- Algorithm Clarity and Completeness

Several algorithmic details remain unclear or undefined, which hinders complete understanding and reproducibility. Algorithm 4 (BasedOnWOAUpdate) references "ApplyRandomMultiRouteOperation" at line 24, but this operation is never formally defined. Similarly, "ApplyRandomOperation" appears in multiple algorithms without specification. While these may be standard VRP neighborhood operators, explicit definitions or references are necessary for clarity.

The feasibility checking and repair mechanisms are mentioned throughout the paper but never formally specified. The phrase "if infeasible then repair" appears frequently, but readers need to understand exactly how capacity violations and time window conflicts are detected and corrected. I recommend adding a dedicated algorithm (perhaps Algorithm 9) that formalizes the feasibility checking and repair procedures, or at minimum, providing a clear textual description in Section 4.

- Presentation and Writing Quality

While the paper is generally well-written, several presentation issues should be addressed. The notation inconsistency throughout the manuscript is distracting. Section 3 introduces "MO-VRPSPDTW" (multi-objective VRPSPDTW) but this term is barely used afterward, with most of the paper referring simply to "VRPSPDTW." Either use the MO prefix consistently if discussing multi-objective optimization, or remove it and focus on the weighted-sum hierarchical objective. Similarly, subscript notation varies between h_{i,j} and h_i in different sections, standardize throughout.

Tables 6-8 mention color coding in the text (gray and yellow shading) to highlight different performance levels, but these colors are not visible in the submitted manuscript. Either ensure the colors appear in the final PDF or remove references to them and rely solely on textual indicators like bold and ** notation.

Section 4.9 (Discussion of the proposed method) largely repeats content from the introduction regarding contributions and novelty. This section could be streamlined to avoid redundancy, perhaps by focusing more on the technical rationale for design decisions rather than restating high-level contributions. The conclusion (Section 7) is relatively brief given the paper's length and could be expanded to better synthesize findings, acknowledge limitations more explicitly, and provide more specific future research directions.

This paper presents valuable research on an important problem and demonstrates competitive experimental results. The hybrid SHO-WOA approach is novel for VRPSPDTW and shows particular promise for clustered customer distributions. The methodology is fundamentally sound, combining complementary metaheuristic strategies with adaptive mechanisms and local improvement.

The required revisions will significantly strengthen the paper. These improvements will transform the contribution from a promising application study into a rigorous methodological advancement with validated performance claims.

With these revisions, the paper will make a solid contribution to both the metaheuristics community and practitioners working on real-world vehicle routing problems. I look forward to seeing the revised manuscript and believe it will be suitable for publication after addressing the points raised above.

**Do you want your identity to be public for this peer review?** For information about this choice, including consent withdrawal, please see our For information about this choice, including consent withdrawal, please see our Privacy Policy .

Reviewer #1: No

Reviewer #2: No

---

## [Author Response · Author response to Decision Letter 1]

19 Jan 2026

We would like to take this opportunity to thank you for your kind comments on the paper “A Parallel Hybrid Meta-heuristic Method for Solving Vehicle Routing Problems with Simultaneous Pickup and Delivery and Time Windows.” We have carefully revised the manuscript and we believe that the revisions have fully addressed the reviewers’ suggestions and concerns.

---

## [Editor Report · Decision Letter 1]

4 Feb 2026

PH-SHOWOA: Parallel Hybrid SHO-WOA for VRPSPDTW

PONE-D-25-56377R1

Dear Dr. Nguyen,

We’re pleased to inform you that your manuscript has been judged scientifically suitable for publication and will be formally accepted for publication once it meets all outstanding technical requirements.

Kind regards,

Nazanin Tajik

Academic Editor

PLOS One
---

## [Editor Report · Acceptance letter]

PONE-D-25-56377R1

PLOS One

Dear Dr. Nguyen,

I'm pleased to inform you that your manuscript has been deemed suitable for publication in PLOS One. Congratulations! Your manuscript is now being handed over to our production team.

Kind regards,

on behalf of

Dr. Nazanin Tajik

Academic Editor

PLOS One